# LIPSCHITZ RECURRENT NEURAL NETWORKS

**N. Benjamin Erichson**
ICSI and UC Berkeley
erichson@berkeley.edu

**Omri Azencot**
Ben-Gurion University
azencot@cs.bgu.ac.il

**Alejandro Queiruga**
Google Research
afq@google.com

**Liam Hodgkinson**
ICSI and UC Berkeley
liam.hodgkinson@berkeley.edu

**Michael W. Mahoney**
ICSI and UC Berkeley
mmahoney@stat.berkeley.edu

## ABSTRACT

Viewing recurrent neural networks (RNNs) as continuous-time dynamical systems, we propose a recurrent unit that describes the hidden state's evolution with two parts: a well-understood linear component plus a Lipschitz nonlinearity. This particular functional form facilitates stability analysis of the long-term behavior of the recurrent unit using tools from nonlinear systems theory. In turn, this enables architectural design decisions before experimentation. Sufficient conditions for global stability of the recurrent unit are obtained, motivating a novel scheme for constructing hidden-to-hidden matrices. Our experiments demonstrate that the Lipschitz RNN can outperform existing recurrent units on a range of benchmark tasks, including computer vision, language modeling and speech prediction tasks. Finally, through Hessian-based analysis we demonstrate that our Lipschitz recurrent unit is more robust with respect to input and parameter perturbations as compared to other continuous-time RNNs.

## 1 INTRODUCTION

Many interesting problems exhibit temporal structures that can be modeled with recurrent neural networks (RNNs), including problems in robotics, system identification, natural language processing, and machine learning control. In contrast to feed-forward neural networks, RNNs consist of one or more recurrent units that are designed to have dynamical (recurrent) properties, thereby enabling them to acquire some form of internal memory. This equips RNNs with the ability to discover and exploit spatiotemporal patterns, such as symmetries and periodic structures (Hinton, 1986). However, RNNs are known to have stability issues and are notoriously difficult to train, most notably due to the vanishing and exploding gradients problem (Bengio et al., 1994; Pascanu et al., 2013).

Several recurrent models deal with the vanishing and exploding gradients issue by restricting the hidden-to-hidden weight matrix to be an element of the orthogonal group (Arjovsky et al., 2016; Wisdom et al., 2016; Mhammedi et al., 2017; Vorontsov et al., 2017; Lezcano-Casado & Martinez-Rubio, 2019). While such an approach is advantageous in maintaining long-range memory, it limits the expressivity of the model. To address this issue, recent work suggested to construct hidden-to-hidden weights which have unit norm eigenvalues and can be nonnormal (Kerg et al., 2019). Another approach for resolving the exploding/vanishing gradient problem has recently been proposed by Kag et al. (2020), who formulate the recurrent units as a differential equation and update the hidden states based on the difference between predicted and previous states.

In this work, we address these challenges by viewing RNNs as dynamical systems whose temporal evolution is governed by an abstract system of differential equations with an external input. The data are formulated in continuous-time where the external input is defined by the function $x = x(t) \in \mathbb{R}^p$, and the target signal is defined as $y = y(t) \in \mathbb{R}^d$. Based on insights from dynamical systems theory, we propose a continuous-time Lipschitz recurrent neural network with the functional form

$$\begin{cases} \dot{h} &= A_{\beta_A, \gamma_A} h + \tanh(W_{\beta_W, \gamma_W} h + Ux + b) \,, & \text{(1a)} \\ y &= Dh \,, & \text{(1b)} \end{cases}$$

where the hidden-to-hidden matrices $A_{\beta,\gamma} \in \mathbb{R}^{N \times N}$ and $W_{\beta,\gamma} \in \mathbb{R}^{N \times N}$ are of the form

$$
\begin{cases}
A_{\beta_A, \gamma_A} &= (1 - \beta_A)(M_A + M_A^T) + \beta_A(M_A - M_A^T) - \gamma_A I & \text{(2a)} \\
W_{\beta_W, \gamma_W} &= (1 - \beta_W)(M_W + M_W^T) + \beta_W(M_W - M_W^T) - \gamma_W I, & \text{(2b)}
\end{cases}
$$

where $\beta_A, \beta_W \in [0, 1]$, $\gamma_A, \gamma_W > 0$ are tunable parameters and $M_A, M_W \in \mathbb{R}^{N \times N}$ are trainable matrices. Here, $h = h(t) \in \mathbb{R}^N$ is a function of time $t$ that represents an internal (hidden) state, and $\dot{h} = \frac{\partial h(t)}{\partial t}$ is its time derivative. The hidden state represents the memory that the system has of its past. The function in Eq. (1) is parameterized by the hidden-to-hidden weight matrices $A \in \mathbb{R}^{N \times N}$ and $W \in \mathbb{R}^{N \times N}$, the input-to-hidden encoder matrix $U \in \mathbb{R}^{N \times p}$, and an offset $b$. The function in Eq. (1b) is parameterized by the hidden-to-output decoder matrix $D \in \mathbb{R}^{d \times N}$. Nonlinearity is introduced via the 1-Lipschitz $\tanh$ activation function. While RNNs that are governed by differential equations with an additive structure have been studied before (Zhang et al., 2014), the specific formulation that we propose in (1) and our theoretical analysis are distinct.

Treating RNNs as dynamical systems enables studying the long-term behavior of the hidden state with tools from stability analysis. From this point of view, an unstable unit presents an exploding gradient problem, while a stable unit has well-behaved gradients over time (Miller & Hardt, 2019). However, a stable recurrent unit can suffer from vanishing gradients, leading to catastrophic forgetting (Hochreiter & Schmidhuber, 1997b). Thus, we opt for a stable model whose dynamics do not (or only slowly do) decay over time. Importantly, stability is also a statement about the robustness of neural units with respect to input perturbations, *i.e.*, stable models are less sensitive to small perturbations compared to unstable models. Recently, Chang et al. (2019) explored the stability of linearized RNNs and provided a *local* stability guarantee based on the Jacobian. In contrast, the particular structure of our unit (1) allows us to obtain guarantees of *global exponential stability* using control theoretical arguments. In turn, the sufficient conditions for global stability motivate a novel symmetric-skew decomposition based scheme for constructing hidden-to-hidden matrices. This scheme alleviates exploding and vanishing gradients, while remaining highly expressive.

In summary, the main contributions of this work are as follows:

- First, in Section 3, using control theoretical arguments in a direct Lyapunov approach, we provide sufficient conditions for *global exponential stability* of the Lipschitz RNN unit (Theorem 1). Global stability is advantageous over local stability results since it guarantees non-exploding gradients regardless of the state. In the special case where $A$ is symmetric, we find that these conditions agree with those in classical theoretical analyses (Lemma 1).

- Next, in Section 4, drawing from our stability analysis, we propose a novel scheme based on the *symmetric-skew decomposition* for constructing hidden-to-hidden matrices. This scheme *mitigates the vanishing and exploding gradients problem*, while obtaining *highly expressive* hidden-to-hidden matrices.

- In Section 6, we show that our Lipschitz RNN has the *ability to outperform state-of-the-art recurrent units* on computer vision, language modeling and speech prediction tasks. Further, our results show that the *higher-order explicit midpoint time integrator* improves the predictive accuracy as compared to using the simpler one-step forward Euler scheme.

- Finally, in Section 7), we study our Lipschitz RNN via the lens of the Hessian and show that it is *robust with respect to parameter perturbations*; we also show that our model is *more robust with respect to input perturbations*, compared to other continuous-time RNNs.

## 2 RELATED WORK

The problem of vanishing and exploding gradients (and stability) have a storied history in the study of RNNs. Below, we summarize two particular approaches to the problem (constructing unitary/orthogonal RNNs and the dynamical systems viewpoint) that have gained significant attention.

**Unitary and orthogonal RNNs.** Unitary recurrent units have received attention recently, largely due to Arjovsky et al. (2016) showing that unitary hidden-to-hidden matrices alleviate the vanishing and exploding gradients problem. Several other unitary and orthogonal models have also been proposed (Wisdom et al., 2016; Mhammedi et al., 2017; Jing et al., 2017; Vorontsov et al., 2017; Jose et al., 2018). While these approaches stabilize the training process of RNNs considerably, they also

limit their expressivity and their prediction accuracy. Further, unitary RNNs are expensive to train, as they typically involve the computation of a matrix inverse at each step of training. Recent work by Lezcano-Casado & Martinez-Rubio (2019) overcame some of these limitations. By leveraging concepts from Riemannian geometry and Lie group theory, their recurrent unit exhibits improved expressivity and predictive accuracy on a range of benchmark tasks while also being efficient to train. Another competitive recurrent design was recently proposed by Kerg et al. (2019). Their approach is based on the Schur decomposition, and it enables the construction of general nonnormal hidden-to-hidden matrices with unit-norm eigenvalues.

**Dynamical systems inspired RNNs.** The continuous time view of RNNs has a long history in the neurodynamics community as it provides higher flexibility and increased interpretability (Pineda, 1988; Pearlmutter, 1995; Zhang et al., 2014). In particular, RNNs that are governed by differential equations with an additive structure have been extensively studied from a theoretical point of view (Funahashi & Nakamura, 1993; Kim et al., 1996; Chow & Li, 2000; Hu & Wang, 2002; Li et al., 2005; Trischler & D'Eleuterio, 2016). See Zhang et al. (2014) for a comprehensive survey of continuous-time RNNs and their stability properties.

Recently, several works have adopted the dynamical systems perspective to alleviate the challenges of training RNNs which are related to the vanishing and exploding gradients problem. For non-sequential data, Ciccone et al. (2018) proposed a negative-definite parameterization for enforcing stability in the RNN during training. Chang et al. (2019) introduced an antisymmetric hidden-to-hidden weight matrix and provided guarantees for local stability. Kag et al. (2020) have proposed a differential equation based formulation for resolving the exploding/vanishing gradients problem by updating the hidden states based on the difference between predicted and previous states. Niu et al. (2019) employed numerical methods for differential equations to study the stability of RNNs.

Another line of recent work has focused on continuous-time models that deal with irregular sampled time-series, missing values and multidimensional time series. Rubanova et al. (2019) and De Brouwer et al. (2019) formulated novel recurrent models based on the theory of differential equations and their discrete integration. Lechner & Hasani (2020) extended these ordinary differential equation (ODE) based models and addresses the issue of vanishing and exploding gradients by designing an ODE-model that is based on the idea of long short-term memory (LSTM). This ODE-LSTM outperforms the continuous-time LSTM (Mei & Eisner, 2017) as well as the GRU-D model (Che et al., 2018) that is based on a gated recurrent unit (GRU).

The link between dynamical systems and models for forecasting sequential data also provides the opportunity to incorporate physical knowledge into the learning process which improves the generalization performance, robustness, and ability to learn with limited data (Chen et al., 2019).

## 3    STABILITY ANALYSIS OF LIPSCHITZ RECURRENT UNITS

One of the key contributions in this work is that we prove that model (1) is *globally exponentially stable* under some mild conditions on $A$ and $W$. Namely, for *any* initial hidden state we can guarantee that our Lipschitz unit converges to an equilibrium if it exists, and therefore, gradients can never explode. We improve upon recent work on stability in recurrent models, which provide only a local analysis, see e.g., (Chang et al., 2019). In fact, global exponential stability is among the strongest notions of stability in nonlinear systems theory, implying all other forms of Lyapunov stability about the equilibrium $h^*$ (Khalil, 2002, Definitions 4.4 and 4.5).

**Definition 1.** *A point $h^*$ is an* equilibrium point *of $\dot{h} = f(h, t)$ if $f(h^*, t) = 0$ for all $t$. Such a point is* globally exponentially stable *if there exists some $C > 0$ and $\lambda > 0$ such that for any choice of initial values $h(0) \in \mathbb{R}^N$,*

$$\|h(t) - h^*\| \leq Ce^{-\lambda t}\|h(0) - h^*\|, \quad \text{for any } t \geq 0. \tag{3}$$

The presence of a Lipschitz nonlinearity in (1) plays an important role in our analysis. While we focus on $\tanh$ in our experiments, our proof is more general and is applicable to models whose nonlinearity $\sigma(\cdot)$ is an $M$-Lipschitz function. Specifically, we consider the general model

$$\dot{h} = Ah + \sigma(Wh + Ux + b), \tag{4}$$

for which we have the following stability result. In the following, we let $\sigma_{\min}$ and $\sigma_{\max}$ denote the smallest and largest singular values of the hidden-to-hidden matrices, respectively.

**Theorem 1.** *Let $h^*$ be an equilibrium point of a differential equation of the form (4) for some $x \in \mathbb{R}^p$. The point $h^*$ is globally exponentially stable if the eigenvalues of $A^{\mathrm{sym}} := \frac{1}{2}(A + A^T)$ are strictly negative, $W$ is non-singular, and either (a) $\sigma_{\min}(A^{\mathrm{sym}}) > M\sigma_{\max}(W)$; or (b) $\sigma$ is monotone non-decreasing, $W + W^T$ is negative definite, and $A^T W + W^T A$ is positive definite.*

The two cases show that global exponential stability is guaranteed if either (a) the matrix $A$ has eigenvalues with real parts sufficiently negative to counteract expanding trajectories in the nonlinearity; or (b) the nonlinearity is monotone, both $A$ and $W$ yield stable linear systems $\dot{u} = Au$, $\dot{v} = Wv$, and $A, W$ have sufficiently similar eigenvectors. In practice, case (b) occasionally holds, but is challenging to ensure without assuming specific structure on $A, W$. Because such assumptions could limit the expressiveness of the model, the next section will develop a tunable formulation for $A$ and $W$ with the capacity to ensure that case (a) holds.

In Appendix A.1, we provide a proof of Theorem 1 using a direct Lyapunov approach. One advantage of this approach is that the driving input $x$ is permitted to evolve in time arbitrarily in the analysis. The proof relies on the classical Kalman-Yakubovich-Popov lemma and circle criterion from control theory — to our knowledge, these tools have not been applied in the modern RNN literature, and we hope our proof can illustrate their value to the community.

In the special case where $A$ is symmetric and $x(t)$ constant, we show that we can also inherit criteria for both local and global stability from a class of well-studied *Cohen–Grossberg–Hopfield models*.

**Lemma 1.** *Suppose that $A$ is symmetric and $W$ is nonsingular. There exists a diagonal matrix $D \in \mathbb{R}^{N \times N}$, and nonsingular matrices $L, V \in \mathbb{R}^{N \times N}$ such that an equilibrium of (4) is (globally exponentially) stable if and only if there is a corresponding (globally exponentially) stable equilibrium for the system*

$$\dot{z} = Dz + L\sigma(Vz + Ux + b). \tag{5}$$

For a thorough review of analyses of (5), see (Zhang et al., 2014). In this special case, the criteria in Theorem 1 coincide with those obtained for the corresponding model (5). However, in practice, we will not choose $A$ to be symmetric.

## 4 SYMMETRIC-SKEW HIDDEN-TO-HIDDEN MATRICES

In this section we propose a novel scheme for constructing hidden-to-hidden matrices. Specifically, based on the successful application of skew-symmetric hidden-to-hidden weights in several recent recurrent architectures, and our stability criteria in Theorem 1, we propose an effective *symmetric-skew decomposition* for hidden matrices. Our decomposition allows for a simple control of the matrix spectrum while retaining its wide expressive range, enabling us to satisfy the spectral constraints derived in the previous section on both $A$ and $W$. The proposed scheme also accounts for the issue of vanishing gradients by reducing the magnitude of large negative eigenvalues.

Recently, several methods used skew-symmetric matrices, *i.e.*, $S + S^T = 0$ to parameterize the recurrent weights $W \in \mathbb{R}^{N \times N}$, see *e.g.*, (Wisdom et al., 2016; Chang et al., 2019). From a stability analysis viewpoint, there are two main advantages for using skew-symmetric weights: these matrices generate the orthogonal group whose elements are isometric maps and thus preserve norms (Lezcano-Casado & Martinez-Rubio, 2019); and the spectrum of skew-symmetric matrices is purely imaginary which simplifies stability analysis (Chang et al., 2019). The main shortcoming of this parametrization is its reduced expressivity, as these matrices have fewer than half of the parameters of a full matrix (Kerg et al., 2019). The latter limiting aspect can be explained from a dynamical systems perspective: skew-symmetric matrices can only describe oscillatory behavior, whereas a matrix whose eigenvalues have nonzero real parts can also encode viable growth and decay information.

To address the expressivity issue, we aim for hidden matrices which on the one hand, allow to control the expansion and shrinkage of their associated trajectories, and on the other hand, will be sampled from a *superset* of the skew-symmetric matrices. Our analysis in Theorem 1 guarantees that Lipschitz recurrent units maintain non-expanding trajectories under mild conditions on $A$ and $W$. Unfortunately, this proposition does not provide any information with respect to the shrinkage of paths. Here, we opt for a system whose expansion and shrinkage can be easily controlled. Formally, the latter requirement is equivalent to designing hidden weights $S$ with small $\mathcal{R}\lambda_i(S)$, $i = 1, 2, \ldots, N$, where $\mathcal{R}(z)$ denotes the real part of $z$. A system of the form (4) whose matrices $A$ and $W$ exhibit

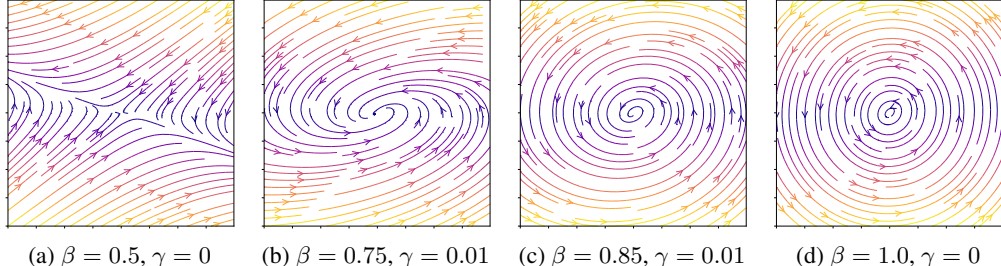

(a) $\beta = 0.5, \gamma = 0$    (b) $\beta = 0.75, \gamma = 0.01$    (c) $\beta = 0.85, \gamma = 0.01$    (d) $\beta = 1.0, \gamma = 0$

Figure 1: Vector fields of hidden states that are governed by Eq. (1) trained for simple pendulum dynamics. In (a), an unstable model is shown. In (b) and (c), it can be seen that we yield models that are asymptotically stable,i.e., all trajectories are attracted by an equilibrium point. In contrast, in (d), a skew-symmetric parameterization leads to a stable model without an attracting equilibrium.

small spectra and satisfy the conditions of Theorem 1, will exhibit dynamics with moderate decay and growth behavior and alleviate the problem of exploding and vanishing gradients. To this end, we propose the following symmetric-skew decomposition for constructing hidden matrices:

$$S_{\beta,\gamma} := (1 - \beta) \cdot (M + M^T) + \beta \cdot (M - M^T) - \gamma I, \tag{6}$$

where $M$ is a weight matrix, and $\beta \in [0.5, 1]$, $\gamma > 0$ are tuning parameters. In the case $(\beta, \gamma) = (1, 0)$, we recover a skew-symmetric matrix, *i.e.*, $S_{1,0} + S_{1,0}^T = 0$. The construction $S_{\beta,\gamma}$ is useful as we can easily bound its spectrum via the parameters $\beta$ and $\gamma$, as we show in the next proposition.

**Proposition 1.** *Let $S_{\beta,\gamma}$ satisfy (6), and let $M^{\mathrm{sym}} = \frac{1}{2}(M + M^T)$. The real parts $\Re\lambda_i(S_{\beta,\gamma})$ of the eigenvalues of $S_{\beta,\gamma}$, as well as the eigenvalues of $S_{\beta,\gamma}^{\mathrm{sym}} = S_{\beta,\gamma} + S_{\beta,\gamma}^T$, lie in the interval*

$$[(1 - \beta)\lambda_{\min}(M^{\mathrm{sym}}) - \gamma, (1 - \beta)\lambda_{\max}(M^{\mathrm{sym}}) - \gamma].$$

A proof is provided in Appendix A.2. We infer that $\beta$ controls the width of the spectrum, while increasing $\gamma$ shifts the spectrum to the left along the real axis, thus enforcing eigenvalues with non-positive real parts. Choosing our hidden-to-hidden matrices to be $A_{\beta_A,\gamma_A}$ and $W_{\beta_W,\gamma_W}$ of the form (6) for different values of $\beta_A$, $\beta_W$ and $\gamma_A$, $\gamma_W$, we can ensure small spectra and satisfy the conditions of Theorem 1 as desired. Note, that different tuning parameters $\beta$ and $\gamma$ affect the stability behavior of the Lipschitz recurrent unit. This is illustrated in Figure 1, where different values for $\beta$ and $\gamma$ are used to construct both $A_{\beta,\gamma}$ and $W_{\beta,\gamma}$ and applied to learning simple pendulum dynamics.

One cannot guarantee that model parameters will remain in the stability region during training. However, we can show that when $\beta$ is taken to be close to one, the eigenvalues of $A_{\beta,\gamma}^{\mathrm{sym}}$ and $W_{\beta,\gamma}^{\mathrm{sym}}$ (which dictate the stability of the RNN) change slowly during training. Let $\Delta_\delta F$ denote the change in a function $F$ depending on the parameters of the RNN (1) after one step of gradient descent with step size $\delta$ with respect to some loss $L(y)$. For a matrix $A$, we let $\lambda_k(A)$ denote the $k$-th singular value of $A$. We have the following lemma.

**Lemma 2.** *As $\beta \to 1^-$, $\max_k |\Delta_\delta \lambda_k(A_{\beta,\gamma}^{\mathrm{sym}})| + \max_k |\Delta_\delta \lambda_k(W_{\beta,\gamma}^{\mathrm{sym}})| = \mathcal{O}(\delta(1 - \beta)^2)$.*

Therefore, provided both the initial and optimal parameters lie within the stability region, the model parameters will remain in the stability region for longer periods of time with high probability as $\beta \to 1$. Further empirical evidence of parameters often remaining in the stability region during training are provided alongside the proof of Lemma 2 in the Appendix (see Figure 5).

## 5    TRAINING CONTINUOUS-TIME LIPSCHITZ RECURRENT UNITS

ODEs such as Eq. (1) can be approximately solved by employing numerical integrators. In scientific computing, numerical integration is a well studied field that provides well understood techniques (LeVeque, 2007). Recent literature has also introduced new approaches which are designed with neural network frameworks in mind (Chen et al., 2018).

To learn the weights $A, W, U$ and $b$, we discretize the continuous model using one step of a numerical integrator between sequence entries. In what follows, a subscript $t$ denotes discrete time indices,

$\Delta t$ represents the time difference between a pair of consecutive data points. Letting $f(h, t) = Ah + \tanh(Wh + Ux(s) + b)$ so that $\dot{h}(t) = f(h, t)$, the exact and approximate solutions for $h_{t+1}$ given $h_t$ are given by

$$h_{t+1} = h_t + \int_t^{t+\Delta t} f(h(s), s)\mathrm{d}s := h_t + \int_t^{t+\Delta t} Ah(s) + \tanh(Wh(s) + Ux(s) + b)\,\mathrm{d}s \quad (7)$$

$$\approx h_t + \Delta t \cdot \mathtt{scheme}\,[f,\, h_t,\, \Delta t]\ , \tag{8}$$

where $\mathtt{scheme}$ represents one step of a numerical integration scheme whose application yields an approximate solution for $\frac{1}{\Delta t}\int_t^{t+\Delta t} f(h(s), s)\mathrm{d}s$ given $h_t$ using one or more evaluations of $f$.

We consider both the explicit (forward) Euler scheme,

$$h_{t+1} = h_t + \Delta t \cdot Ah_t + \Delta t \cdot \tanh(z_t), \tag{9}$$

as well as the midpoint method which is a two-stage explicit Runge-Kutta scheme (RK2),

$$h_{t+1} = h_t + \Delta t \cdot A\tilde{h} + \Delta t \cdot \tanh(W\tilde{h} + Ux_t + b), \tag{10}$$

where $\tilde{h} = h_t + \Delta t/2 \cdot Ah_t + \Delta t/2 \cdot \tanh(z_t)$ is an intermediate hidden state. The RK2 scheme can potentially improve the performance since the scheme is more accurate, however, this scheme also requires twice as many function evaluations as compared to the forward Euler scheme. Given a $\beta$ and $\gamma$ that yields a globally exponentially stable continuous model, $\Delta t$ can always be chosen so that the model remains in the stability region of forward Euler and RK2 (LeVeque, 2007).

## 6 EMPIRICAL EVALUATION

In this section, we evaluate the performance of the Lipschitz RNN and compare it to other state-of-the-art methods. The model is applied to ordered and permuted pixel-by-pixel MNIST classification, as well as to audio data using the TIMIT dataset. We show the sensitivity with respect to to random initialization in Appendix B. Appendix B also contains additional results for: pixel-by-pixel CIFAR-10 and a noise-padded version of CIFAR-10; as well as for character level and word level prediction using the Penn Tree Bank (PTB) dataset. All of these tasks require that the recurrent unit learns long-term dependencies: that is, the hidden-to-hidden matrices need to have sufficient memory to remember information from far in the past.

### 6.1 ORDERED AND PERMUTED PIXEL-BY-PIXEL MNIST

The pixel-by-pixel MNIST task tests long range dependency by sequentially presenting 784 pixels to the recurrent unit, *i.e.*, the RNN processes one pixel at a time (Le et al., 2015). At the end of the

Table 1: Evaluation accuracy on ordered and permuted pixel-by-pixel MNIST.

| Name | ordered | permuted | N | # params |
|---|---|---|---|---|
| LSTM baseline by (Arjovsky et al., 2016) | 97.3% | 92.7% | 128 | ≈68K |
| MomentumLSTM (Nguyen et al., 2020) | 99.1% | 94.7% | 256 | ≈270K |
| Unitary RNN (Arjovsky et al., 2016) | 95.1% | 91.4% | 512 | ≈9K |
| Full Capacity Unitary RNN (Wisdom et al., 2016) | 96.9% | 94.1% | 512 | ≈270K |
| Soft orth. RNN (Vorontsov et al., 2017) | 94.1% | 91.4% | 128 | ≈18K |
| Kronecker RNN (Jose et al., 2018) | 96.4% | 94.5% | 512 | ≈11K |
| Antisymmteric RNN (Chang et al., 2019) | 98.0% | 95.8% | 128 | ≈10K |
| Incremental RNN (Kag et al., 2020) | 98.1% | 95.6% | 128 | ≈4K/8K |
| Exponential RNN (Lezcano-Casado & Martinez-Rubio, 2019) | 98.4% | 96.2% | 360 | ≈69K |
| Sequential NAIS-Net (Ciccone et al., 2018) | 94.3% | 90.8% | 128 | ≈18K |
| Lipschitz RNN using Euler (ours) | 99.0% | 94.2% | 64 | ≈9K |
| Lipschitz RNN using RK2 (ours) | 99.1% | 94.2% | 64 | ≈9K |
| Lipschitz RNN using Euler (ours) | **99.4%** | **96.3%** | 128 | ≈34K |
| Lipschitz RNN using RK2 (ours) | 99.3% | 96.2% | 128 | ≈34K |

Table 2: Evaluation on TIMIT using 1 layer models. The mean squared error (MSE) is computes the distance between the predicted and actual log-magnitudes of each predicted frame in the sequence.

| Name | val. MSE | test MSE | N | # params |
|---|---|---|---|---|
| LSTM (Helfrich et al., 2018) | 13.66 | 12.62 | 158 | ≈200K |
| LSTM (Nguyen et al., 2020) | 9.33 | 9.37 | 158 | ≈200K |
| MomentumLSTM (Nguyen et al., 2020) | 5.86 | 5.87 | 158 | ≈200K |
| SRLSTM (Nguyen et al., 2020) | 5.81 | 5.83 | 158 | ≈200K |
| Full-capacity Unitary RNN (Wisdom et al., 2016) | 14.41 | 14.45 | 256 | ≈200K |
| Cayley RNN (Helfrich et al., 2018) | 7.97 | 7.36 | 425 | ≈200K |
| Exponential RNN (Lezcano-Casado & Martinez-Rubio, 2019) | 5.52 | 5.48 | 425 | ≈200K |
| Lipschitz RNN using Euler (ours) | 2.95 | 2.82 | 256 | ≈198K |
| Lipschitz RNN using RK2 (ours) | **2.86** | **2.76** | 256 | ≈198K |

sequence, the learned hidden state is used to predict the class membership probability of the input image. This task requires that the RNN has a sufficient long-term memory in order to discriminate between different classes. A more challenging variation to this task is to operate on a fixed random permutation of the input sequence.

Table 1 provides a summary of our results. The Lipschitz RNN, with hidden dimension of $N = 128$ and trained with the forward Euler and RK2 scheme, achieves $99.4\%$ and $99.3\%$ accuracy on the ordered pixel-by-pixel MNIST task. For the permuted task, the model trained with forward Euler achieves $96.3\%$ accuracy, whereas the model trained with RK2 achieves $96.2\%$ accuracy. Hence, our Lipschitz recurrent unit outperforms state-of-the-art RNNs on both tasks and is competitive even when a hidden dimension of $N = 64$ is used, however, it can be seen that a larger unit with more capacity is advantageous for the permuted task. Our results show that we significantly outperform the Antisymmetric RNN (Chang et al., 2019) on the ordered tasks, while using fewer weights. That shows that the antisymmetric weight paramterization is limiting the expressivity of the recurrent unit. The exponential RNN is the next most competitive model, yet this model requires a larger hidden-to-hidden unit to perform well on the two considered tasks.

## 6.2 TIMIT

Next, we consider the TIMIT dataset (Garofolo, 1993) to study the capabilities of the Lipschitz RNN for speech prediction using audio data. For our experiments, we used the publicly available implementation of this task by Lezcano-Casado & Martinez-Rubio (2019). This implementation applies the preprocessing steps suggested by Wisdom et al. (2016): (i) downsample each audio sequence to 8kHz; (ii) process the downsampled sequences with a short-time Fourier transform using a Hann window of 256 samples and a window hop of 128 samples; and (iii) normalize the log-magnitude of the Fourier amplitudes. We obtain a set of frames that each have 129 complex-valued Fourier amplitudes and the task is to predict the log-magnitude of future frames. To compare our results with those of other models, we used the common train / validation / test split: 3690 utterances from 462 speakers for training, 192 utterances for validation, and 400 utterances for testing.

Table 2 lists the results for the Lipschitz recurrent unit as well as for several benchmark models. It can be seen that the Lipschitz RNN outperforms other state-of-the-art models for a fixed number of parameters ($\approx$ 200K). In particular, LSTMs do not perform well on this task, however, the recently proposed momentum based LSTMs (Nguyen et al., 2020) have improvemed performance. Interestingly, the RK2 scheme leads to a better performance since this scheme provides more accurate approximations for the intermediate states.

## 7 ROBUSTNESS WITH RESPECT TO PERTURBATIONS

An important consideration beyond accuracy is robustness with respect to input and parameter perturbations. We consider a Hessian-based analysis and noise-response analysis of different continuous-time recurrent units and train the models on MNIST. Here, we reshape each MNIST thumbnail into sequences of length 98 so that each input has dimension $x \in \mathbb{R}^8$. We consider this

simpler problem so that all models obtain roughly the same training loss. Here we use stochastic gradient decent (SGD) with momentum to train the models.

Eigenanalysis of the Hessian provides a tool for studying various aspects of neural networks (Hochreiter & Schmidhuber, 1997a; Sagun et al., 2017; Ghorbani et al., 2019). Here, we study the Hessian $H$ spectrum with respect to the model parameters of the recurrent unit using Py-Hessian (Yao et al., 2019). The Hessian provides us with insights about the curvature of the loss function $\mathcal{L}$. This is because the Hessian is defined as the derivatives of the gradients, and thus the Hessian eigenvalues describe the change in the gradient of $\mathcal{L}$ as we take an infinitesimal step into a given direction. The eigenvectors span the (local) surface of the loss function at a given point, and the corresponding eigenvalue determines the curvature in the direction of the eigenvectors. This means that larger eigenvalues indicate a larger curvature, *i.e.*, greater sensitivity, and the sign of the eigenvalues determines whether the curvature will be positive or negative.

To demonstrate the advantage of the additional linear term and our weight parameterization, we compare the Lipschitz RNN to two other continuous-time recurrent units. First, we consider a simple neural ODE RNN (Rubanova et al., 2019) that takes the form

$$\dot{h} = \tanh(Wh + Ux + b), \qquad y = Dh, \qquad (11)$$

where $W$ is a simple hidden-to-hidden matrix. As a second model we consider the antisymmetric RNN (Chang et al., 2019), that takes the same form as (11), but uses a skew-symmetric scheme to parameterize the hidden-to-hidden matrix as $W := (M - M^T) - \gamma I$, where $M$ is a trainable weight matrix and $\gamma$ is a tunable parameter.

Table 3 reports the largest eigenvalue $\lambda_{\max}(H)$ and the trace of the Hessian $\mathrm{tr}(H)$. The largest eigenvalue being smaller indicates that our Lipschitz RNN found a flatter minimum, as compared to the simple neural ODE and Antisymmetric RNN. It is known that such flat minima can be perturbed without significantly changing the loss value (Hochreiter & Schmidhuber, 1997a). Table 3 also reports the condition number $\kappa(H) := \frac{\lambda_{\max}(H)}{\lambda_{\min}(H)}$ of the Hessian. The condition number $\kappa(H)$ provides a measure for the spread of the eigenvalues of the Hessian. It is known that first-order methods can slow down in situations where $\kappa$ is large (Bottou & Bousquet, 2008). The condition number and trace of our Lipshitz RNN being smaller also indicates improved robustness properties.

Next, we study the sensitivity of the response $y_T$ at time $T$ in terms of the test accuracy with respect to a sequence of perturbed inputs $\{\tilde{x}_1, \ldots, \tilde{x}_T\} \in \mathbb{R}^8$. We consider three different perturbations. The results for the artificially constructed perturbations are presented in Table 3, showing that the Lipschitz RNN is more resilient to adversarial perturbation. Here, we have considered the projected gradient decent (PGD) (Goodfellow et al., 2014) method with $l_\infty$, and the DeepFool

Table 3: Summary of Hessian-based robustness metrics and resilience to adversarial attacks.

| Model | PGD | DF$_2$ | DF$_\infty$ | $\lambda_{\max}(H)$ | $\mathrm{tr}(H)$ | $\kappa(H)$ |
|---|---|---|---|---|---|---|
| Neural ODE RNN | 88.5% | 69.6% | 44.5% | 0.30 | 4.7 | 37.6 |
| Antisymmetric RNN | 84.7% | 83.4% | 44.3% | 0.24 | 4.8 | 35.5 |
| Lipschitz RNN (ours) | **93.0%** | **89.2%** | **54.1%** | **0.14** | **3.1** | **23.2** |

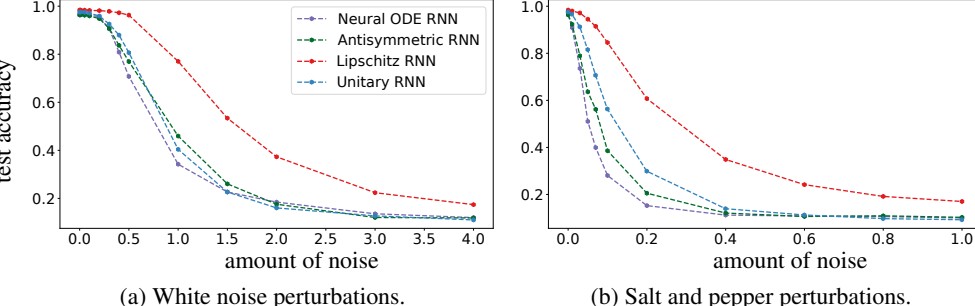

(a) White noise perturbations.     (b) Salt and pepper perturbations.

Figure 2: Sensitivity with respect to different input perturbations.

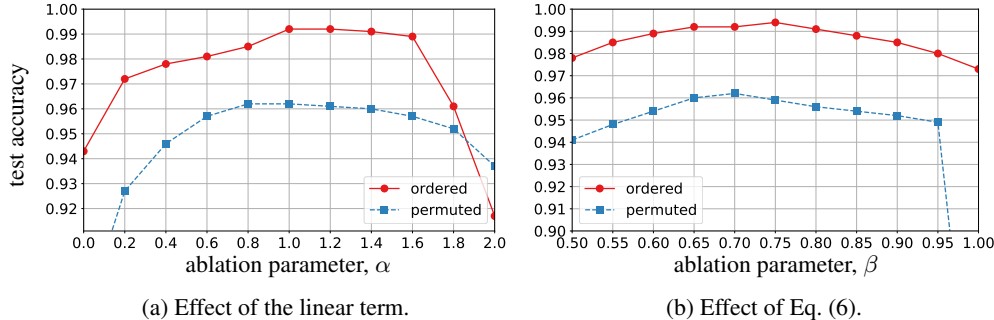

(a) Effect of the linear term.

(b) Effect of Eq. (6).

Figure 3: The ablation study examines the effect of the linear term $Ah$ (in (a)) and the importance of the Skew-Symmetric Decomposition for constructing the hidden-to-hidden matrices (in (b)).

method (Moosavi-Dezfooli et al., 2016) with $l_2$ and $l_\infty$ norm ball perturbations. We construct the adversarial examples with full access to the models, using 7 iterations. The step size for PGD is set to 0.01.

Further, Figure 2 shows the results for white noise and salt and pepper noise. It can be seen that the Lipschitz unit is less sensitive to input perturbations, as compared to the simple neural ODE RNN, and the antisymmetric RNN. In addition, we also show the results for an unitary RNN here.

## 7.1 ABLATION STUDY

The performance of the Lipschitz recurrent unit is due to two main innovations: (i) the additional linear term; and (ii) the scheme for constructing the hidden-to-hidden matrices $A$ and $W$ in Eq. (6). Thus, we investigate the effect of both innovations, while keeping all other conditions fixed. More concretely, we consider the following ablation recurrent unit

$$h_{t+1} = h_t + \alpha \cdot \epsilon \cdot Ah_t + \epsilon \cdot \tanh(z_t), \quad \text{with} \quad z_t = Wh_t + Ux_t + b, \tag{12}$$

where $\alpha$ controls the effect of the linear hidden unit. Both $A$ and $W$ depend on the parameters $\beta, \gamma$.

Figure 3a studies the effect of the linear hidden unit, with $\beta = 0.65$ for the ordered task and $\beta = 0.8$ for the permuted task. In both cases we use $\gamma = 0.001$. It can be seen that the test accuracies of both the ordered and permuted pixel-by-pixel MNIST tasks clearly depend on the linear hidden unit. For $\alpha = 0$, our models reduces to simple neural ODE recurrent units (Eq. (11)). The recurrent unit degenerates for $\alpha > 1.6$, since the external input is superimposed by the hidden state. Figure 3b studies the effect of the hidden-to-hidden matrices with respect to $\beta$. It can be seen that $\beta = \{0.65, 0.70\}$ achieves peak performance for the ordered task, and $\beta = \{0.8, 0.85\}$ does so for the permuted task. Note that $\beta = 1.0$ recovers an skew-symmetric hidden-to-hidden matrix.

## 8 CONCLUSION

Viewing RNNs as continuous-time dynamical systems with input, we have proposed a new Lipschitz recurrent unit that excels on a range of benchmark tasks. The special structure of the recurrent unit allows us to obtain guarantees of global exponential stability using control theoretical arguments. In turn, the insights from this analysis motivated the symmetric-skew decomposition scheme for constructing hidden-to-hidden matrices, which mitigates the vanishing and exploding gradients problem. Due to the nice stability properties of the Lipschitz recurrent unit, we also obtain a model that is more robust with respect to input and parameter perturbations as compared to other continuous-time units. This behavior is also reflected by the Hessian analysis of the model. We expect that the improved robustness will make Lipschitz RNNs more reliable for sensitive applications. The theoretical results for our symmetric-skew decomposition of parameterizing hidden-to-hidden matrices also directly extend to the convolutional setting. Future work will explore this extension and study the potential advantages of these more parsimonious hidden-to-hidden matrices in combination with our parameterization in practice. Research code is shared via github.com/erichson/LipschitzRNN.

ACKNOWLEDGMENTS

We would like to thank Ed H. Chi for fruitful discussions about physics-informed machine learning and the Antisymmetric RNN. We are grateful to the generous support from Amazon AWS and Google Cloud. NBE and MWM would like to acknowledge IARPA (contract W911NF20C0035), NSF, ONR and CLTC for providing partial support of this work. Our conclusions do not necessarily reflect the position or the policy of our sponsors, and no official endorsement should be inferred.

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

# A PROOFS

## A.1 PROOFS OF THEOREM 1 AND LEMMA 1

There are numerous ways that one can analyze the global stability of (4) through the related model (5), many of which are discussed in Zhang et al. (2014). Instead, here we shall conduct a direct approach and avoid appealing to diagonalization in order to obtain cleaner conditions, and a more straightforward proof that readily applies in the time-inhomogeneous setting.

Our method of choice relies on Lyapunov arguments summarized in the following theorem, which can be found as (Khalil, 2002, Theorem 4.10). For more details on related Lyapunov theory, see also Hahn (1967); Sastry (2013).

**Theorem 2.** *An equilibrium $h^*$ for $\dot{h} = f(t, h)$ is globally exponentially stable if there exists a continuously differentiable function $V : [0, \infty) \times \mathbb{R}^N \to [0, \infty)$ such that for all $h \in \mathbb{R}^N$ and $t \geq 0$,*

$$k_1 \|h - h^*\|^\alpha \leq V(t, h) \leq k_2 \|h - h^*\|^\alpha, \quad and \quad \frac{\partial V}{\partial t} + \frac{\partial V}{\partial h} \leq -k_3 \|h - h^*\|^\alpha,$$

*for some constants $k_1, k_2, k_3, \alpha > 0$. and $\dot{V}(h) < 0$ for $h \neq h^*$.*

To simplify matters, we shall choose a Lyapunov function $V : \mathbb{R}^N \to [0, \infty)$ that is independent of time. The most common type of Lyapunov function satisfying the conditions of Theorem 2 is of the form $V(h) = (h - h^*)^T P(h - h^*)$, where $P$ is a positive definite matrix. One need only show that $\dot{V}(h) \leq -(h - h^*)^T Q(h - h^*)$ for some other positive definite matrix $Q$ to guarantee global exponential stability.

The construction of the Lyapunov function $V$ that satisfies the conditions of Theorem 2 is accomplished using the Kalman-Yakubovich-Popov lemma, which is a statement regarding *strictly positive real transfer functions*. We use the following definition, equivalent to other standard definitions by (Khalil, 2002, Lemma 6.1).

**Definition 2.** *A function $G : \mathbb{C} \to \mathbb{C}^{N \times N}$ is* strictly positive real *if it satisfies the following:*

(i) *The poles of $G(s)$ have negative real parts.*

(ii) *$G(i\omega) + G(-i\omega)^T$ is positive definite for all $\omega \in \mathbb{R}$, where $i = \sqrt{-1}$.*

(iii) *Either $G(\infty) + G(\infty)^T$ is positive definite or it is positive semidefinite and $\lim_{\omega \to \infty} \omega^2 M^T[G(i\omega) + G(-i\omega)^T]M$ is positive definite for any $N \times (N - q)$ full-rank matrix $M$ such that $M^T[G(\infty) + G(\infty)^T]M = 0$, where $q = \mathrm{rank}[G(\infty) + G(\infty)^T]$.*

The following is presented in (Khalil, 2002, Lemma 6.3).

**Lemma 3** (Kalman-Yakubovich-Popov). *Let $A, W : \mathbb{R}^N \to \mathbb{R}^N$ be full-rank square matrices. There exists a symmetric positive-definite matrix $P$ and matrices $L, U$ and a constant $\epsilon > 0$ such that*

$$PA + A^T P = -L^T L - \epsilon P$$
$$P = L^T U - W^T$$
$$U^T U = 0,$$

*if and only if the transfer function $G(s) = W(sI - A)^{-1}$ is strictly positive real. In this case, we may take $\epsilon = 2\mu$, where $\mu > 0$ is chosen so that $G(s - \mu)$ remains strictly positive real.*

A shorter proof for case (a) is available to us through the (multivariable) *circle criterion* — the following theorem is a corollary of (Khalil, 2002, Theorem 7.1) suitable for our purposes.

**Theorem 3** (Circle Criterion). *The system of differential equations*

$$\dot{h} = Ah + \psi(t, Wh)$$

*is globally exponentially stable towards an equilibrium at the origin if $\|\psi(t, y)\| \leq M\|y\|$ for some $M > 0$ and $Z(s) = [I + MG(s)][I - MG(s)]^{-1}$ is strictly positive real, where $G(s) = W(sI - A)^{-1}$.*

Both the Kalman-Yakubovich-Popov lemma and the circle criterion are classical results in control theory, and are typically discussed in the setting of feedback systems (Khalil, 2002, Chapter 6, 7). Our presentation here is less general than the complete formulation, but makes clearer the connection to RNNs. With these tools, we state our proof of Theorem 1.

*Proof of Theorem 1.* To begin, we shall center the differential equation about the equilibrium. By assumption, there exists $h^*$ such that $Ah^* = -\sigma(Wh^* + Ux(t) + b)$. Letting $\bar{h} = h - h^*$, we find that

$$
\begin{aligned}
\dot{\bar{h}} &= Ah + \sigma(Wh + Ux(t) + b) \\
&= A\bar{h} + Ah^* + \sigma(W\bar{h} + Wh^* + Ux(t) + b) \\
&= A\bar{h} + \sigma(W\bar{h} + Wh^* + Ux(t) + b) - \sigma(Wh^* + Ux(t) + b).
\end{aligned}
\tag{13}
$$

It will suffice to show that (13) is globally exponentially stable at the origin.

Let us begin with case (a). The proof follows arguments analogous to (Khalil, 2002, Example 7.1). Let $G(s) = W(A - sI)^{-1}$ denote the transfer function for the system (13). Letting

$$
\psi(t, x) = \sigma(x + Wh^* + Ux(t) + b) - \sigma(Wh^* + Ux(t) + b),
$$

since $\sigma$ is $M$-Lipschitz, we know that $\|\psi(t, x)\| \leq M\|x\|$ for any $x \in \mathbb{R}^N$. Therefore, let $Z(s) = [I + MG(s)][I - MG(s)]^{-1}$ denote the transfer function in the circle criterion. Our objective is to show that $Z(s)$ is strictly positive real — by Theorem 3, this will guarantee the desired global exponential stability of (4). First, we need to show that the poles of $Z(s)$ have negative real parts. This can only occur when $G(s)$ itself has poles or $I - MG(s)$ is singular. The former case occurs precisely where $A - sI$ is singular, which occurs when $s$ is an eigenvalue of $A$. Since $A + A^T$ is assumed to be negative definite, $A$ must have eigenvalues with negative real part by Lemma 4, and so the poles of $G(s)$ also have negative real parts. The latter case is more difficult to treat. First, since $\sigma_{\max}(AB) \leq \sigma_{\max}(A)\sigma_{\max}(B)$ and $\sigma_{\max}(B^{-1}) = \sigma_{\min}(B)^{-1}$,

$$
\sigma_{\max}(G(s)) \leq \frac{\sigma_{\max}(W)}{\sigma_{\min}(A - sI)}.
\tag{14}
$$

Therefore, we observe that

$$
\begin{aligned}
\sigma_{\min}(I - MG(s)) &\geq 1 - \sigma_{\max}(MG(s)) \\
&\geq 1 - M\sigma_{\max}(G(s)) \\
&\geq 1 - \frac{M\sigma_{\max}(W)}{\sigma_{\min}(A - sI)}.
\end{aligned}
$$

From the Fan-Hoffman inequality (Bhatia, 2013, Proposition III.5.1), we have that

$$
\sigma_{\min}(A - sI) = \sigma_{\min}(sI - A) \geq \lambda_{\min}\left(\Re(s)I - \frac{A + A^T}{2}\right) = \Re(s) + \lambda_{\min}\left(-\frac{A + A^T}{2}\right),
$$

and since $A + A^T$ is negative definite, for any $s$ with $\Re(s) \geq 0$,

$$
\sigma_{\min}(A - sI) \geq \Re(s) + \sigma_{\min}\left(\frac{A + A^T}{2}\right) \geq \sigma_{\min}(A^{\mathrm{sym}}).
\tag{15}
$$

Since $\sigma_{\min}(A^{\mathrm{sym}}) > M\sigma_{\max}(W)$, it follows that $\sigma_{\min}(I - MG(s)) > 0$ whenever $s$ has non-negative real part, and so the poles of $Z(s)$ must have negative real parts.

Next, we need to show that $Z(i\omega) + Z(-i\omega)^T$ is positive definite for all $\omega \in \mathbb{R}$. Observe that

$$
\begin{aligned}
Z(i\omega) + Z(-i\omega)^T &= [I + MG(i\omega)][I - MG(i\omega)]^{-1} + [I - MG(-i\omega)^T]^{-1}[I + MG(-i\omega)^T] \\
&= 2[I - MG(-i\omega)^T]^{-1}[I - M^2G(-i\omega)^T G(i\omega)][I - MG(i\omega)]^{-1}.
\end{aligned}
$$

From Sylvester's law of inertia, we may infer that $Z(i\omega) + Z(-i\omega)^T$ is positive definite if and only if $I + Y_\omega$ is positive definite, where $Y_\omega = M^2G(-i\omega)^T G(i\omega)$. If we can show that the eigenvalues of $Y_\omega$ lie strictly within the unit circle, that is, $\sigma_{\max}(Y_\omega) < 1$ for all $\omega \in \mathbb{R}$, then $I + Y_\omega$ will necessarily be positive definite. From (14) and (15), we may verify that

$$
\sup_{\omega \in \mathbb{R}} \sigma_{\max}(G(i\omega)) \leq \sup_{\omega \in \mathbb{R}} \frac{\sigma_{\max}(W)}{\sigma_{\min}(A - i\omega I)} \leq \frac{\sigma_{\max}(W)}{\sigma_{\min}(A^{\mathrm{sym}})}.
$$

Therefore,

$$\sigma_{\max}(Y_\omega) \le M^2 \sigma_{\max}(G(-i\omega)^T)\sigma_{\max}(G(i\omega)) \le \left(\frac{M\sigma_{\max}(W)}{\sigma_{\min}(A^{\mathrm{sym}})}\right)^2 < 1,$$

by assumption. Finally, since $Z(\infty) + Z(\infty)^T = 2I$ is positive definite, $Z(s)$ is strictly positive real and Theorem 3 applies.

Now, consider case (b). The proof proceeds in two steps. First, we verify that the transfer function $G(s) = W(A - sI)^{-1}$ satisfies the conditions of the Kalman-Yakubovich-Popov lemma. Then, using the matrices $P, L, U$, and the constant $\epsilon$ inferred from the lemma, a Lyapunov function is constructed which satisfies the conditions of Theorem 2, guaranteeing global exponential stability. Once again, condition (i) of Lemma 3 is straightforward to verify: $G(s)$ exhibits poles when $s$ is an eigenvalue of $A$, and so the poles of $G(s)$ also have negative real parts. Furthermore, condition (iii) is easily satisfied with $M = I$ since $G(\infty) + G(\infty)^T = 0$. To show that condition (ii) holds, observe that for any $\omega \in \mathbb{R}$, letting $A^{-T} = (A^{-1})^T$ for brevity,

$$G(i\omega) + G(-i\omega)^T = W(A - i\omega I)^{-1} + (A + i\omega I)^{-T}W^T$$
$$= (A + i\omega I)^{-T}[(A + i\omega I)^T W + W^T(A - i\omega I)](A - i\omega I)^{-1}.$$

Since the inner matrix factor is Hermitian, Sylvester's law of inertia implies that $G(i\omega) + G(-i\omega)^T$ is positive definite if and only if

$$B_\omega \coloneqq (A + i\omega I)^T W + W^T(A - i\omega I).$$

is positive definite. Since $B_\omega$ is a Hermitian matrix, it has real eigenvalues, with minimal eigenvalue given by the infimum of the Rayleigh quotient:

$$\lambda_{\min}(B_\omega) = \inf_{\|v\|=1} v^T B_\omega v$$
$$= \inf_{\|v\|=1} v^T(A^T W + W^T A)v + i\omega v^T(W - W^T)v$$
$$= \inf_{\|v\|=1} v^T(A^T W + W^T A)v$$
$$= \lambda_{\min}(A^T W + W^T A).$$

By assumption, $A^T W + W^T A$ has strictly positive eigenvalues, and hence $B_\omega$ and $G(i\omega) + G(-i\omega)^T$ are positive definite. Therefore, Lemma 3 applies, and we obtain matrices $P, L, U$ and a constant $\epsilon > 0$ with the corresponding properties.

Now we may construct our Lyapunov function $V$. Let $v = W\bar{h}$ and

$$u(t) = \sigma(v(t) + Wh^* + Ux(t) + b) - \sigma(Wh^* + Ux(t) + b),$$

so that $\dot{\bar{h}} = A\bar{h} + u$. Since $\sigma$ is monotone non-decreasing, $\sigma(x) - \sigma(y) \ge 0$ for any $x \ge y$. This implies that for each $i = 1, \dots, N$, $v_i$ and $u_i$ have the same sign. In particular, $v^T u \ge 0$. Now, let $V(h) = h^T P h$ be our Lyapunov function, noting that $V$ is independent of $t$. Taking the derivative of the Lyapunov function over (13) and using the properties of $P, L, U, \epsilon$,

$$\dot{V}(\bar{h}) = \bar{h}^T P \dot{\bar{h}} + \dot{\bar{h}}^T P \bar{h}$$
$$= \bar{h}^T(PA + A^T P)\bar{h} + 2\bar{h}^T P u$$
$$= \bar{h}^T(-L^T L - \epsilon P)\bar{h} + 2\bar{h}^T(L^T U - W^T)u$$
$$= -(L\bar{h})^T(L\bar{h}) + (L\bar{h})^T Uu + (Uu)^T(L\bar{h}) - u^T U^T Uu - 2v^T u$$
$$= -(L\bar{h} + Uu)^T(L\bar{h} + Uu) - \epsilon\bar{h}^T P\bar{h} - 2v^T u.$$

Since $v^T u \ge 0$ and $(L\bar{h} + Uu)^T(L\bar{h} + Uu) \ge 0$, it follows that $\dot{V}(\bar{h}) \le -\epsilon\lambda_{\min}(P)\|h\|^2$, and hence global exponential stability follows from Theorem 2 and positive-definiteness of $P$. $\qquad\square$

To finish off discussion regarding the results from Sec. 3, we provide a quick proof of Lemma 1 using a simple diagonalization argument.

*Proof of Lemma 1.* Since $A$ is symmetric and real-valued, by (Horn & Johnson, 2012, Theorem 4.1.5), there exists an orthogonal matrix $P$ and a real diagonal matrix $D$ such that $A = PDP^T$. Letting $z = P^T h$ where $h$ satisfies (4), since $h = Pz$, we see that

$$\dot{z} = P^T PDP^T h + P^T \sigma(Wh + Ux + b)$$
$$= Dz + P^T \sigma(WPz + Ux + b).$$

Therefore, $z$ satisfies (5) with $L = P^T$ and $V = WP$, both of which are nonsingular by orthogonality of $P$. By the same argument, for any equilibrium $h^*$, taking $z^* = P^T h^*$,

$$Dz^* + P^T \sigma(WPz^* + Ux + b) = P^T(PDP^T h^* + \sigma(Wh^* + Ux + b))$$
$$= P^T(Ah^* + \sigma(Wh^* + Ux + b)) = 0,$$

implying that $z^*$ is an equilibrium of (5). Furthermore, since

$$\|z - z^*\|^2 = (P^T h - P^T h^*)^T (P^T h - P^T h^*)$$
$$= (h - h^*)^T PP^T (h - h^*) = \|h - h^*\|^2,$$

from orthogonality of $P$. Because every form of Lyapunov stability, both local and global, including global exponential stability, depend only on the norm $\|h - h^*\|$ (Khalil, 2002, Definitions 4.4 and 4.5), $h^*$ is stable under any of these forms if and only if $z^*$ is also stable. $\qquad\square$

We remark that the proof of Lemma 1 can extend to matrices $A$ which have real eigenvalues and are diagonalizable. These attributes are implied for symmetric matrices. However, they can be difficult to ensure in practice for nonsymmetric matrices without imposing difficult structural constraints.

## A.2 PROOF OF PROPOSITION 1

The proof of Proposition 1 relies on the following lemma, which we also have made use of several times throughout this work.

**Lemma 4.** *For any matrix $A \in \mathbb{R}^{N \times N}$, the real parts of the eigenvalues $\Re\lambda_i(A)$ are contained in the interval $[\lambda_{\min}(A^{\mathrm{sym}}), \lambda_{\max}(A^{\mathrm{sym}})]$, where $A^{\mathrm{sym}} = \frac{1}{2}(A + A^T)$.*

*Proof.* Recall by the min-max theorem, for $\langle u, v \rangle = u^* v$, where $u^*$ is the conjugate transpose of $u$, the upper and lower eigenvalues of $A + A^T$ satisfy

$$\lambda_{\min}(A + A^T) = \inf_{v \in \mathbb{C}^N, \, \|v\|=1} \langle v, (A + A^T)v \rangle = \inf_{v \in \mathbb{C}^N, \, \|v\|=1} \langle v, Av \rangle + \langle Av, v \rangle,$$

$$\lambda_{\max}(A + A^T) = \sup_{v \in \mathbb{C}^N, \, \|v\|=1} \langle v, (A + A^T)v \rangle = \sup_{v \in \mathbb{C}^N, \, \|v\|=1} \langle v, Av \rangle + \langle Av, v \rangle.$$

Let $\lambda_i(A) = u + i\omega$ be an eigenvalue of $A$ with corresponding eigenvector $v$ satisfying $\|v\| = 1$. Since $Av = (u + i\omega)v$,

$$\langle v, Av \rangle + \langle Av, v \rangle = \langle v, Av \rangle + \overline{\langle v, Av \rangle} = 2\Re\langle v, Av \rangle = 2u\|v\|^2 = 2u.$$

Hence, $\lambda_{\min}(A + A^T) \leq u \leq \lambda_{\max}(A + A^T)$. $\qquad\square$

*Proof of Proposition 1.* By construction, $S_{\beta,\gamma}^{\mathrm{sym}} = S_{\beta,\gamma} + S_{\beta,\gamma}^T = (1 - \beta)M^{\mathrm{sym}} - \gamma I$, and so from Lemma 4, both the real parts $\Re\lambda_i(S_{\beta,\gamma})$ of the eigenvalues of $S_{\beta,\gamma}$ as well as the eigenvalues of $S_{\beta,\gamma}^{\mathrm{sym}}$ lie in the interval

$$[\lambda_{\min}(S_{\beta,\gamma}^{\mathrm{sym}}), \lambda_{\max}(S_{\beta,\gamma}^{\mathrm{sym}})] = [\lambda_{\min}((1 - \beta)M^{\mathrm{sym}} - \gamma I), \lambda_{\max}((1 - \beta)M^{\mathrm{sym}} - \gamma I)].$$

If $\beta < 1$, for any eigenvalue $\lambda$ of $S_{\beta,\gamma}^{\mathrm{sym}}$ with corresponding eigenvector $v$,

$$(1 - \beta)M^{\mathrm{sym}}v - \gamma v = \lambda v, \quad \text{and so} \quad M^{\mathrm{sym}}v = \frac{\lambda + \gamma}{1 - \beta}v$$

implying that $\frac{\lambda+\gamma}{1-\beta}$ is an eigenvalue of $M^{\mathrm{sym}}$, and therefore contained in $[\lambda_{\min}(M^{\mathrm{sym}}), \lambda_{\max}(M^{\mathrm{sym}})]$. In particular, we find that

$$[\lambda_{\min}(S_{\beta,\gamma}^{\mathrm{sym}}), \lambda_{\max}(S_{\beta,\gamma}^{\mathrm{sym}})] \subseteq [(1 - \beta)\lambda_{\min}(M^{\mathrm{sym}}) - \gamma, (1 - \beta)\lambda_{\max}(M^{\mathrm{sym}})], \qquad (16)$$

as required. Finally, if $\beta = 1$, then (16) still holds, since both intervals collapse to the single point $\{-\gamma\}$. $\qquad\square$

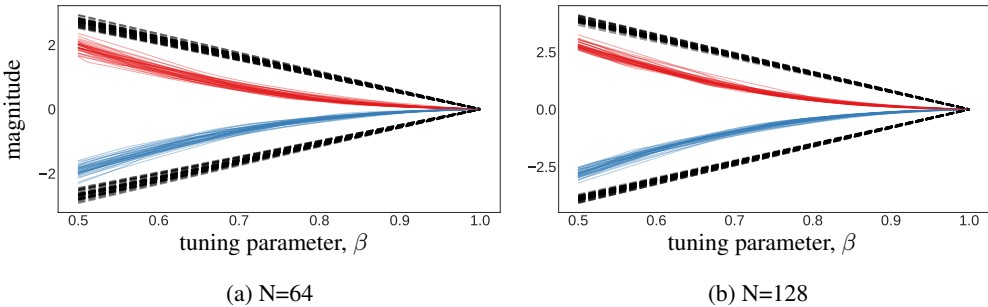

(a) N=64             (b) N=128

Figure 4: Empirical evaluation of the theoretical bounds (16). The red lines track the largest real part and the blue lines track the smallest real part of the eigenvalues of the hidden-to-hidden matrix $A_\beta$. Each line corresponds to a different hidden-to-hidden matrix of dimension $N = 64$ in (a) and $N = 128$ in (b). The dashed black lines indicate the theoretical bound for each trial.

Figure 4 illustrates the effect of $\beta$ onto the eigenvalues of $A_{\beta,\gamma}$ with the largest and smallest real parts. It can be seen, both empirically and theoretically, that the real part of the eigenvalues converges towards zero as $\beta$ tends towards one, *i.e.*, we yield a skew-symmetric matrix with purely imaginary eigenvalues in the limit. Thus, for a sufficiently large parameter $\beta$ we yield a system that approximately preserves an "energy" for a limited time-horizon

$$\mathcal{R}\lambda_i(A_{\beta,\gamma}) \approx 0, \quad \text{for} \quad i = 1, 2, \ldots, N. \tag{17}$$

### A.3    PROOF OF LEMMA 2

First, it follows from Gronwall's inequality that the norm of the final hidden state $\|h(T)\|$ is bounded uniformly in $\beta$. From Weyl's inequalities and the definition of $A_{\beta,\gamma}$,

$$\max_k |\Delta_\delta \lambda_k(A_{\beta,\gamma}^{\text{sym}})| \le \|\Delta_\delta A_{\beta,\gamma}^{\text{sym}}\| = (1-\beta)\|\Delta_\delta M_A^{\text{sym}}\|.$$

By the chain rule, for each element $M_A^{ij}$ of the matrix $M_A$,

$$\frac{\partial L}{\partial M_A^{ij}} = \frac{\partial L}{\partial y(T)} \frac{\partial y(T)}{\partial h(T)} \frac{\partial h(T)}{\partial M_A^{ij}} = \frac{\partial L}{\partial y(T)} D \frac{\partial h(T)}{\partial M_A^{ij}}.$$

Now, for any collection of parameters $\theta_i$,

$$\frac{d}{dt}\sum_i \frac{\partial h}{\partial \theta_i} = A \sum_i \frac{\partial h}{\partial \theta_i} + \sum_i \frac{\partial A}{\partial \theta_i} h + \text{sech}^2\left(Wh + Ux + b\right)\left(W \sum_i \frac{\partial h}{\partial \theta_i} + \sum_i \frac{\partial W}{\partial \theta_i} h\right),$$

and from Gronwall's inequality,

$$\left\|\sum_i \frac{\partial h(T)}{\partial \theta_i}\right\| \le \left(\left\|\sum_i \frac{\partial A_{\beta,\gamma}}{\partial \theta_i}\right\| + \left\|\sum_i \frac{\partial W_{\beta,\gamma}}{\partial \theta_i}\right\|\right) \|h\| e^{(\|A_{\beta,\gamma}\| + \|W_{\beta,\gamma}\|)T}.$$

Since $\Delta_\delta M_A^{\text{sym}} = \delta \frac{\partial L}{\partial M_A} + \delta\left(\frac{\partial L}{\partial M_A}\right)^T$,

$$\|\Delta_\delta M_A^{\text{sym}}\| \le \|\Delta_\delta M_A^{\text{sym}}\|_F$$

$$\le \delta\sqrt{\sum_{i,j}\left(\frac{\partial L}{\partial M_A^{ij}} + \frac{\partial L}{\partial M_A^{ij}}\right)^2}$$

$$\le \delta \left\|\frac{\partial L}{\partial y}\right\| \|D\| \|h\| e^{(\|A_{\beta,\gamma}\| + \|W_{\beta,\gamma}\|)T}\sqrt{\sum_{i,j}\left\|\frac{\partial A_{\beta,\gamma}}{\partial M_A^{ij}} + \frac{\partial A_{\beta,\gamma}}{\partial M_A^{ji}}\right\|^2}.$$

Since $\frac{\partial(M_A h)}{\partial M_A^{ij}} = \frac{\partial(M_A^T h)}{\partial M_A^{ji}}$, it follows that

$$\frac{\partial A_{\beta,\gamma}}{\partial M_A^{ij}} + \frac{\partial A_{\beta,\gamma}}{\partial M_A^{ji}} = 2(1-\beta)\left(\frac{\partial(M_A h)}{\partial M_A^{ij}} + \frac{\partial(M_A^T h)}{\partial M_A^{ji}}\right),$$

and so $\|\Delta_\delta M_A^{\text{sym}}\| = \mathcal{O}(\delta(1-\beta))$, and therefore $\max_k |\Delta_\delta \sigma_k(A_{\beta,\gamma}^{\text{sym}})| = \mathcal{O}(\delta(1-\beta)^2)$. Similarly, for the matrix $M_W$,

$$\max_k \left| \Delta_\delta \lambda_k(W_{\beta,\gamma}^{\text{sym}}) \right| \leq (1-\beta) \|\Delta_\delta M_W^{\text{sym}}\|$$

$$\leq \delta(1-\beta) \left\| \frac{\partial L}{\partial y} \right\| \|D\| \|h\| e^{(\|A_{\beta,\gamma}\| + \|W_{\beta,\gamma}\|)T} \sqrt{\sum_{i,j} \left\| \frac{\partial W_{\beta,\gamma}}{\partial M_W^{ij}} + \frac{\partial W_{\beta,\gamma}}{\partial M_W^{ji}} \right\|^2}$$

$$= 2\delta(1-\beta)^2 \left\| \frac{\partial L}{\partial y} \right\| \|D\| \|h\| e^{(\|A_{\beta,\gamma}\| + \|W_{\beta,\gamma}\|)T} \sqrt{\sum_{i,j} \left( \frac{\partial (M_W h)}{\partial M_W^{ij}} + \frac{\partial (M_W^T h)}{\partial M_W^{ji}} \right)^2},$$

and hence $\max_k |\Delta_\delta \lambda_k(W_{\beta,\gamma}^{\text{sym}})| = \mathcal{O}(\delta(1-\beta)^2)$. $\square$

In Figure 5, we plot the most positive real part of the eigenvalues of $A_{\beta,\gamma}$ and $W_{\beta,\gamma}$ during training for the ordered MNIST task. As $\beta$ increases, the eigenvalues change less during training, remaining in the stability region provided by case (b) of Theorem 1 for more of the training time.

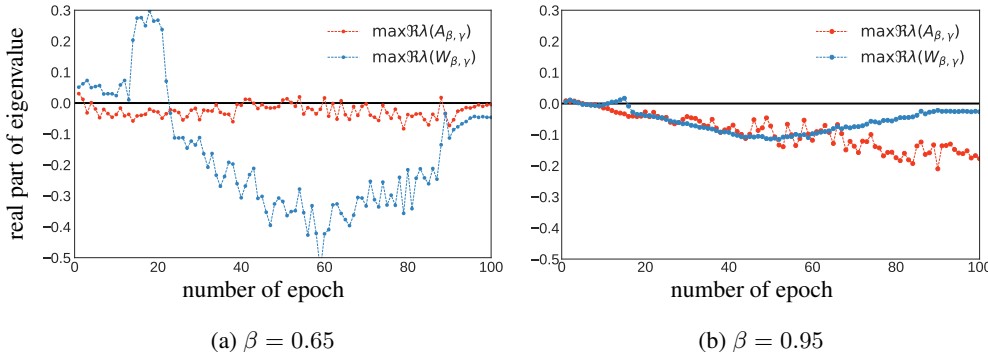

(a) $\beta = 0.65$            (b) $\beta = 0.95$

Figure 5: The red lines track the largest real part of the eigenvalues of the hidden-to-hidden matrix $A_{\beta,\gamma}$ and the blue lines track the largest real part of the eigenvalues of $W_{\beta,\gamma}$. We show results for two models trained on the ordered MNIST task with varying $\beta$.

## B  ADDITIONAL EXPERIMENTS

### B.1  SENSITIVITY TO RANDOM INITIALIZATION FOR MNIST AND TIMIT

The hidden matrices are initialized by sampling weights from the normal distribution $\mathcal{N}(0, \sigma)$, where $\sigma$ is the variance, which can be treated as a tuning parameter. In our experiments we typically chose a small $\sigma$; see the Table 8 for details. To show that the Lipschitz RNN is insensitive to random initialization, we have trained each model with 10 different seeds. Table 4 shows the maximum, average and minimum values obtained for each task. Note that higher values indicate better performance on the ordered and permuted MNIST tasks, while lower values indicate better performance on the TIMIT task.

### B.2  ORDERED PIXEL-BY-PIXEL AND NOISE-PADDED CIFAR-10

The pixel-by-pixel CIFAR-10 benchmark problem that has recently been proposed by (Chang et al., 2019). This task is similar to the pixel-by-pixel MNIST task, yet more challenging due to the increased sequence length and the more difficult classification problem. Similar to MNIST, we flatten the CIFAR-10 images to construct a sequence of length $1024$ in scanline order, where each element of the sequence consists of three pixels (one from each channel).

A variation of this problem is the noise-padded CIFAR-10 problem (Chang et al., 2019), where we consider each row of an image as input at time step $t$. The rows from each channel are stacked so that we obtain an input of dimension $x \in \mathbb{R}^{96}$. Then, after the 32 time step which process the 32

Table 4: Sensitivity to random initialization evaluated over 10 runs.

| Solver | Task | Minimum | Average | Maximum | N | # params |
|--------|------|---------|---------|---------|---|----------|
| Euler | ordered MNIST | 98.9% | 99.0% | 99.0% | 64 | ≈9K |
| RK2 | ordered MNIST | 98.9% | 99.0% | 99.1% | 64 | ≈9K |
| Euler | ordered MNIST | 99.0% | 99.2% | 99.4% | 128 | ≈34K |
| RK2 | ordered MNIST | 98.9% | 99.1% | 99.3% | 128 | ≈34K |
| Euler | permuted MNIST | 93.5% | 93.8% | 94.2% | 64 | ≈9K |
| RK2 | permuted MNIST | 93.5% | 93.9% | 94.2% | 64 | ≈9K |
| Euler | permuted MNIST | 95.6% | 95.9% | 96.3% | 128 | ≈34K |
| RK2 | permuted MNIST | 95.4% | 95.8% | 96.2% | 128 | ≈34K |
| Euler | TIMIT (test MSE) | 2.82 | 2.98 | 3.10 | 256 | ≈198K |
| RK2 | TIMIT (test MSE) | 2.76 | 2.81 | 2.84 | 256 | ≈198K |

Table 5: Evaluation accuracy on pixel-by-pixel CIFAR-10 and noise padded CIFAR-10.

| Name | ordered | noise padded | N | # params |
|------|---------|--------------|---|----------|
| LSTM baseline by (Chang et al., 2019) | 59.7% | 11.6% | 128 | 69K |
| Antisymmetric RNN (Chang et al., 2019) | 58.7% | 48.3% | 256 | 36K |
| Incremental RNN (Kag et al., 2020) | - | 54.5% | 128 | - |
| Lipschitz RNN using Euler (ours) | 60.5% | 57.4% | 128 | 34K/46K |
| Lipschitz RNN using RK2 (ours) | 60.3% | 57.3% | 128 | 34K/46K |
| Lipschitz RNN using Euler (ours) | **64.2%** | **59.0%** | 256 | 134K/158K |
| Lipschitz RNN using RK2 (ours) | 64.2% | 58.9% | 256 | 134K/158K |

row, we start to feed the recurrent unit with independent standard Gaussian noise for 968 time steps. At the final point in $T = 1000$, we use the learned hidden state for classification. This problem is challenging because only the first 32 time steps contain signals. Thus, the recurrent unit needs to recall information from the beginning of the process.

Table 5 provides a summary of our results. Our Lipschitz recurrent unit outperforms both the incremental RNN (Kag et al., 2020) and the antisymmetric RNN (Chang et al., 2019) by a significant margin. This impressively demonstrates that the Lipschitz unit enables the stable propagation of signals over long time horizons.

## B.3   PENN TREE BANK (PTB)

### B.3.1   CHARACTER LEVEL PREDICTION

Next, we consider a character level language modeling task using the Penn Treebank Corpus (PTB) (Marcus et al., 1993). Specifically, this task studies how well a model can predict the next character in a sequence of text. The dataset is composed of a train / validation / test set, where 5017K characters are used for training, 393K characters are used for validation and 442K characters are used for testing. For our experiments, we used the publicly available implementation of this task by Kerg et al. (2019), which computes the performance in terms of mean bits per character (BPC).

Table 6 shows the results for back-propagation through time (BPTT) over 150 and 300 time steps, respectively. The Lipschitz RNN performs slightly better then the exponential RNN and the non-normal RNN on this task. (Kerg et al., 2019) notes that orthogonal hidden-to-hidden matrices are not particular well-suited for this task. Thus, it is not surprising that the Lipschitz unit has a small advantage here.

For comparison, we have also tested the Antisymmetric RNN (Chang et al., 2019) on this task. The performance of this unit is considerably weaker as compared to our Lipschitz unit. This suggests that the Lipschitz RNN is more expressive and improves the propagation of meaningful signals over longer time scales.

Table 6: Evaluation accuracy on PTB for character-level prediction for different sequence lengths $T$. The * indicate results that were adopted from Kerg et al. (2019).

| Name | $T_{PTB} = 150$ | $T_{PTB} = 300$ | # params |
|---|---|---|---|
| RNN baseline by (Arjovsky et al., 2016) | 2.89 | 2.90 | $\approx$1.32M |
| RNN-orth (Henaff et al., 2016) (*) | 1.62 | 1.66 | $\approx$1.32M |
| EURNN (Jing et al., 2017) (*) | 1.61 | 1.62 | $\approx$1.32M |
| Exponential RNN (Lezcano-Casado & Martinez-Rubio, 2019) (*) | 1.49 | 1.52 | $\approx$1.32M |
| Non-normal RNN (Kerg et al., 2019) | 1.47 | 1.49 | $\approx$1.32M |
| Antisymmteric RNN | 1.60 | 1.64 | $\approx$1.32M |
| Lipschitz RNN using Euler (ours) | **1.43** | **1.46** | $\approx$1.32M |

### B.3.2 WORD-LEVEL PREDICTION

In addition to character-level prediction, we also consider word-level prediction using the PTB corpus. For comparison with other state-of-the-art units, we consider the setup by Kusupati et al. (2018), who use a sequence length of 300. Table 7 shows results for back-propagation through time (BPTT) over 300 time steps. The Lipschitz RNN performs slightly better than the other RNNs on this task and the baseline LSTM for the test perplexity metric reported by Kusupati et al. (2018).

Table 7: Evaluation accuracy on PTB for word-level prediction. The * indicate results adopted from Kusupati et al. (2018). Note that here the parameters for the hidden-to-hidden units are reported.

| Name | validation perplexity | test perplexity | N | # params |
|---|---|---|---|---|
| LSTM (*) | - | 117.41 | - | 210K |
| SpectralRNN (*) | - | 130.20 | - | 24.8K |
| FastRNN (*) | - | 127.76 | - | 52.5K |
| FastGRNN-LSQ (*) | - | 115.92 | - | 52.5K |
| FastGRNN (*) | - | 116.11 | - | 52.5K |
| Incremental RNN (Kag et al., 2020) | - | 115.71 | - | 29.5K |
| Lipschitz RNN using Euler (ours) | 124.55 | **115.36** | 160 | 50K |

## C TUNING PARAMETERS

For tuning we utilized a standard training procedure using a non-exhaustive random search within the following plausible ranges for the our weight parameterization $\beta = 0.65, 0.7, 0.75, 0.8$, $\gamma = [0.001, 1.0]$. For Adam we explored learning rates between 0.001 and 0.005, and for SGD we considered 0.1. For the step size we explored values in the range 0.001 to 1.0. We did not perform an automated grid search and thus expect that the models can be further fine-tuned.

The tuning parameters for the different tasks that we have considered are summarized in Table 8.

For pixel-by-pixel MNIST and CIFAR-10, we use Adam for minimizing the objective. We train all our models for 100 epochs, with scheduled learning rate decays at epochs $\{90\}$. We do not use gradient clipping during training. Figure 6 shows the test accuracy curves for our Lipschitz RNN for the ordered and permuted MNIST classification tasks.

For TIMIT we use Adam with default parameters for minimizing the objective. We also tried Adam using betas (0.0, 0.9) as well as RMSprop with $\alpha = 0.9$, however, Adam with default values worked best in our experiments. We train the model for 1200 epochs without learning-rate decay. Similar to Kerg et al. (2019) we train our model with gradient clipping, however, we observed that the performance of our model is relatively insensitive to the clipping value.

For the character level prediction task, we use Adam with default parameters for minimizing the objective, while we use RMSprop with $\alpha = 0.9$ for the word level prediction task. We train the model for 200 epochs for the character-level task, and for 500 epochs for the word-level task.

Table 8: Tuning parameters used for our experimental results and the performance evaluated with 12 different seed values for the parameter initialization of the model.

| Name | N | lr | decay | $\beta$ | $\gamma_a$ | $\gamma_w$ | $\epsilon$ | $\sigma$ |
|------|---|-----|-------|---------|-----------|-----------|-----------|---------|
| Ordered MNIST | 64 | 0.003 | 0.1 | 0.75 | 0.001 | 0.001 | 0.03 | $0.1/64$ |
| Ordered MNIST | 128 | 0.003 | 0.1 | 0.75 | 0.001 | 0.001 | 0.03 | $0.1/128$ |
| Permuted MNIST | 64 | 0.0035 | 0.1 | 0.75 | 0.001 | 0.001 | 0.03 | $0.1/128$ |
| Permuted MNIST | 128 | 0.0035 | 0.1 | 0.75 | 0.001 | 0.001 | 0.03 | $0.1/128$ |
| Ordered CIFAR10 | 256 | 0.1 | 0.2 | 0.65 | 0.001 | 0.001 | 0.01 | $6/256$ |
| Noise-padded CIFAR10 | 256 | 0.1 | 0.2 | 0.75 | 0.001 | 0.001 | 0.01 | $6/256$ |
| TIMIT | 256 | 0.001 | - | 0.8 | 0.8 | 0.001 | 0.9 | $12/256$ |
| PTB character-level 150 | 750 | 0.005 | - | 0.8 | 0.5 | 0.001 | 0.1 | $12/256$ |
| PTB character-level 300 | 750 | 0.005 | - | 0.8 | 0.5 | 0.001 | 0.1 | $12/256$ |
| PTB word-level | 160 | 0.1 | - | 0.8 | 0.9 | 0.001 | 0.01 | $10/256$ |

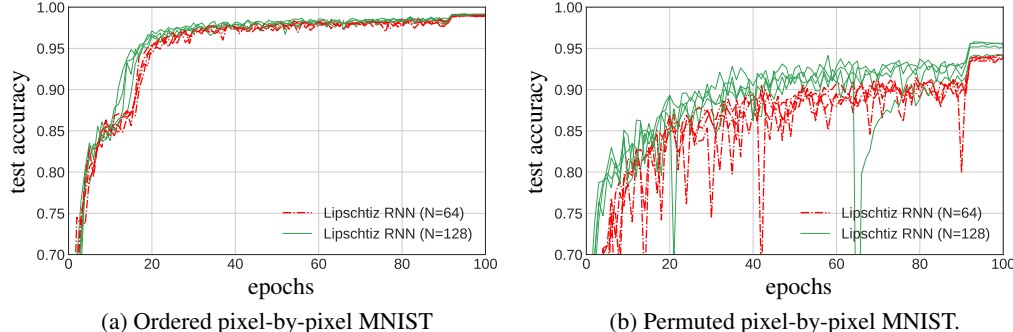

(a) Ordered pixel-by-pixel MNIST          (b) Permuted pixel-by-pixel MNIST.

Figure 6: Test accuracy for the Lipschitz RNN for different classification tasks.

