# OpenReview forum: "Lipschitz Recurrent Neural Networks"
_ICLR.cc/2021/Conference — ICLR 2021 Poster_

### Official Review · AnonReviewer2 · 2020-10-22
**Solid and well-done paper, but quite close to previous work, and with some caveats from a dynamical systems perspective.**

**Rating:** 7
**Confidence:** 4

**Review:**

Considering a continuous time RNN with Lipschitz-continuous nonlinearity, the authors formulate sufficient conditions on the parameter matrices for the network to be globally stable, in the sense of a globally attracting fixed point. They provide a specific parameterization for the hidden-to-hidden weight matrices to control global stability and error gradients, consisting of a weighted combination of a symmetric and a skew-symmetric matrix (and some diagonal offset). The authors discuss numerical integration by forward-Euler and RK2, and thoroughly benchmark their approach against a large set of other state-of-the-art RNNs on various tasks including versions of MNIST and TIMIT.  Finally, they highlight improved stability of their RNN against parameter and input perturbations.

What I like about this paper is that it provides a solid theoretical basis and a principled, insightful parameterization that let’s one control separately the size of the real and the imaginary parts of the eigenvalues of A and W. The paper contains a number of interesting thoughts, and a really extensive comparison to other state-of-the-art models on several benchmarks.

In general, I feel however that this work is relatively close to the 2019 ICLR paper by Chang et al.; in several ways it feels like a more or less straightforward extension of this previous work.

It also remains a bit unclear to me how it’s ensured in practice that the matrices obey to the required conditions in the training process. Sect. 5 is not really about training, but just about numerically solving the ODE. From Appendix C it seems the scalar parameters $\beta, \gamma$ controlling the influence of the symmetric vs. skew-symmetric parts and the offset are not learned at all but just fixed after grid-search? The component matrices B, C, on the other hand it seems are not restricted at all but just initialized such that the theoretical conditions are likely, but not necessarily, met? This seems somewhat unsatisfying as there are in fact no guarantees that the global stability conditions will be met in practice, and tuning the model may require (potentially extensive) meta-parameter search?

Another drawback in my mind is that enforcing global fixed point dynamics is quite restrictive. For instance, this rules out cycles and many other interesting dynamics in the model’s intrinsic behavior. Apparently, from the authors’ empirical tests, this seems not to be required for solving this particular set of tasks. Which is somewhat puzzling to me as it appears this assumption should strongly curtail the model’s expressiveness.

In the tables and figures I missed statistics. No standard errors or confidence bands were provided, or how many runs were performed. If it’s all from a single run, can I be certain the numbers are not just lucky draws?

Nevertheless, given the overall convincing and extensive empirical results, I’m slightly leaning toward acceptance.

Minor issues:
- The authors sell the additional linear term in their RNN as a novelty, while in fact it’s rather standard in continuous RNN (older papers by Barak Pearlmutter, Song & Wang 2016, arxiv.org/abs/2006.02427, arxiv.org/abs/1910.03471)
- RK2 is generally not sufficient for more involved (stiff) dynamical problems; so the reason it works well here may lie in the fact that the model’s intrinsic dynamic is indeed very simple
- What is an unstable unit? I guess the authors mean that the RNN is not globally stable?
- Sect. 2,  dynamical systems inspired RNN: It may be important to note that formulating a RNN as ODE does not solve the exploding/vanishing gradient or stability problem per se (nor is it immediately clear to me why it should actually make it easier).
- Theorem 1: The $\sigma$ refer to the matrix eigenvalues in this case?
- What is a superset of skew-symmetric matrices?
- Second-to-last pg. of Sect. 7 was unclear to me, i.e. what exactly was done and evaluated here, maybe cos I’m not familiar with some of the cited methods.

---

> ### Author Response · Authors · 2020-11-17
> **A fair and insightful assessment — we can address some of the claimed limitations.**
>
> We thank the reviewer for the detailed feedback and insights. We are pleased to see that the reviewer likes the paper and recognizes the contributions to the literature. There are a few comments raised that we would like to further comment on and hope to address.
>
> First, we acknowledge that our work took primary inspiration from Chang et al. (2019). We found their particular choice of parameterization to be far too restrictive in practice, and our primary objective was to improve upon their ideas through a less restrictive parameterization that aligns with theory.
>
> **The reviewer raises three good points in particular that we would like to comment on:**
>
> (1) it cannot be ensured that the hidden-to-hidden matrices remain in the “stable” region during training;
>
> (2) enforcing global stability could be too restrictive; and
>
> (3) the success of RK2 in practice seems to suggest that the resulting ODE is not stiff, and therefore cannot have differentials with eigenvalues with large negative real parts.
>
>
> All are astute observations. We agree with (2), strong global stability is indeed problematic and to be avoided. Indeed, strong global stability is linked to the vanishing gradient problem. However, when the eigenvalues have only small negative real parts (sometimes called the edge of chaos), approximate cycles and other interesting behavior can still occur. Strong global stability would reveal stiff ODEs, so from (3) we know this does not occur during training. Indeed, regarding (1), once beta and gamma have been appropriately chosen such that beta is relatively close to one and gamma ensures we lie in the stability region initially, experiments show that the real parts of the eigenvalues remain small and negative. Our newly added lemma also sheds some light on this, showing that taking beta close to one ensures that the real parts of the eigenvalues cannot change by too much, even though the training and test losses do change significantly. In the extreme case where $\beta = 1$ , there is no risk of leaving the stability region, and we revert to the work of Chang et al. (2019) but this significantly impacts expressivity.
>
> Achieving a good balance is still a matter of manually tuning these parameters. One could try to also optimize these parameters, but this proves too difficult in practice, and we have found it unnecessary. However, the strategy mentioned above prevents any search from becoming too extensive.
>
> **Regarding the other comments:**
>
> * Standard errors over multiple runs will be provided; we can confirm the runs are not simply lucky.
> * An RNN is considered unstable if it is not globally stable
> * The continuous-time approach offers (arguably) simpler analysis and often larger stability regions.
> * The superset of skew-symmetric matrices considered here is the set of all matrices.
>
> **References**
>
> Chang, B., Chen, M., Haber, E., & Chi, E. H. (2019). AntisymmetricRNN: A Dynamical System View on Recurrent Neural Networks. In International Conference on Learning Representations.

---

> > ### Comment · AnonReviewer2 · 2020-11-22
> > **Role of the symmetric matrix part (beyond antisym. RNN) still a bit unclear to me**
> >
> > I thank the authors for taking my comments seriously, and for their attempts to resolve them.
> > Before this goes into the final discussion stage, I would like to give the authors another opportunity to respond to some of my remaining concerns.
> >
> > The authors argue that when initializing parameters within the stability region, they will likely remain there throughout training. They show two specific examples (Fig. 5) where this is (nearly) the case (unfortunately no statistics, % of cases), and state a lemma that provides a reason why this is likely to be the case. While this may not be completely satisfying, I’m largely happy with this part.
> >
> > A point that remains in my mind and that is perhaps difficult to argue away is the closeness of this work to the paper by Chang et al. First, the same conditions that ensure stability ($\beta$ → 1) seem to push the network into a regime where it doesn’t differ too much anymore from the formulation by Chang et al. Second, whether there is the claimed huge increase in expressiveness may be debatable: The authors’ RNN always converges to the same fixed point in the absence of inputs, while the Chang RNN can express (true) complex cycles if I’m not mistaken. The authors’ are right, however, that their system can still exhibit damped oscillations (spiral points). Although the experimental results appear to confirm better performance of the authors’ RNN compared to Antisymmetric RNN, sometimes this may be due to the larger number of parameters, and Antisymmetric RNN were not provided as a baseline in about half of the experiments. I also wondered whether the set of tasks used by the authors really challenges expressiveness and long-memory properties that much, since these are all tasks where potentially meaningful inputs were continuously provided (at each time step). So I guess it’s not so clear how long the times really are across which long memory is required. What would happen if one had something like the multiplication problem where meaningful inputs are separated by long delays filled with mere noise?
> >
> > I’m aware I’m pushing things a bit here and this argumentation may not seem completely fair also in light of other work on this topic. However, the point I’m trying to get at is that I’m still missing a bit a stronger demonstration or argument that incorporating the symmetric matrix part is a major step beyond the Chang work (esp. when $\beta$ is close to1). If parameters $\alpha$ and $\beta$ were trainable (yet still ensuring stability) and could adapt the symmetric and anti-symmetric contributions to the problem at hand, that in my mind would be more convincing. I'm aware though this is likely beyond the present paper, but some other more specific example or formal argument may help as well.

---

> > > ### Author Response · Authors · 2020-11-23
> > > **Response Part I**
> > >
> > > We would like to thank the reviewer for taking the time to reply to our response. Your concerns are justified, but we strongly believe that we can demonstrate the advantage of our model.
> > > We believe there is a fundamental difference in philosophy between our approach and that of Chang et al. and previous works in the RNN literature. As we pointed out in our response to Reviewer 3, our method does not project the hidden-to-hidden weight matrices onto a subspace where stability and/or non-vanishing gradients are guaranteed in the infinite-time horizon. This is a deliberate choice, because guaranteeing these properties is not always beneficial, as it limits expressivity. Our parameterization for *any* $\beta \notin \{0,1\}$ does not restrict the space of available hidden-to-hidden weight matrices in the infinite-time horizon. Instead, it acts as an aid for training, like adding a few extra hyperparameters to the optimizer. In our new lemma, we show that beta directly affects how much the eigenvalues change during training, and that the training process is unlikely to rapidly move into an area with catastrophic vanishing/exploding gradient behavior when beta is large. In practice, we do not need to take $\beta$ to be very close to one for this to occur. Indeed, Figure 3(b) shows that taking $\beta$ to be too close to one dramatically reduces performance. The precise results in the right half of the figure are tabulated below.
> > >
> > > |                                    | $\beta$       | $\gamma$  |  Accuracy   | N                | #parms         |
> > > |----------------------- --|:-------------:|:-------------:|:-------------:|:-------------:|:--------------:|
> > > | Lipschitz RNN            |   0.75          |  0.001         |   99.4%       |  128            |  34K             |
> > > | Lipschitz RNN            |   0.90          |  0.001         |   98.6           |  128            |  34K             |
> > > | Lipschitz RNN            |   0.95          |  0.001         |   98.2           |  128            |  34K             |
> > > | Lipschitz RNN            |   1.0            |  0.001         |   97.3           |  128            |  34K             |
> > >
> > > Recalling that $\beta = 1$ (without the linear term) is the antisymmetric RNN, we find that the antisymmetric RNN encounters a catastrophic loss in expressivity, which results in a major drop in performance. From our experiments (e.g. Table 1), we know that this cannot be attributed to having a smaller number of parameters, because even when we account for this, our method still achieves much better accuracy. Our practical contribution is that by not restricting the process to remain in the stability region a la antisymmetric RNN, one can achieve greatly improved performance, and we show how this can be achieved in a fairly reliable way.
> > >
> > > **"The Chang RNN can express (true) complex cycles if I’m not mistaken"**
> > >
> > > In practice, Chang et al. train their models with a small amount of diffusion, i.e., $W = M - M^T - \gamma I$ to ensure that the model is in the stability region of the forward Euler scheme. A good choice of $\gamma$ is very difficult to find. Putting this aside, however, in theory, the antisymmetric RNN is designed such that it can only learn perfect cycles. If the optimal choice of hidden-to-hidden weight matrices exhibits perfect cycles also, then there is nothing preventing our formulation from also reproducing this effect. However, we have found that good choices of weights tend to exhibit slight dampening or expanding behavior. By allowing for this behavior, we can achieve much better performance.

---

> > > > ### Author Response · Authors · 2020-11-23
> > > > **Response Part II**
> > > >
> > > > **More detailed comparison to the Antisymmetric RNN**
> > > >
> > > > We strongly believe that our contributions are *strictly* more general than Chang et al. from both theoretical and practical perspectives. On the theory side, we prove that our model exhibits global stability, whereas Chang et al. only show local stability results. As for the practical aspect, our model consistently outperforms Chang et al. in *all* cases. In particular, we borrowed the results reported in Chang et al. for the MNIST and CIFAR-10 datasets, and the result by Kerg et al. (2019) for the PTB character level dataset.
> > > > Unfortunately, to the best of our knowledge, there is no existing publicly available implementation of the Antisymmetric RNN model. Therefore, we tested the Antisymmetric RNN model using our implementation which we prepared by following the details in Chang et al. However,  we were not able to reproduce the results reported in Chang et al. for the MNIST and CIFAR-10 tests, even after an extensive search over the hyperparameters domain. We list below some of the results we obtained when fine-tuning our in-house implementation of Chang et al. work. Moreover, we extensively experimented with the Antisymmetric model on the TIMIT dataset, please see below. For a similar number of parameters, our model and other state-of-the-art baselines substantially outperform the Antisymmetric RNN. Thus, we decided to *not* include the results of the antisymmetric RNN for TIMIT as it is unfortunately not competitive enough. We are happy to include these results in our revision.
> > > >
> > > > Results for ordered and permuted pixel-by-pixel MNIST.
> > > >
> > > > |                                    | $\gamma$  | ordered       | permuted     | N                | #parms        |
> > > > |--------------------------|:-------------:|:-------------:|:-------------:|:-------------:|:--------------:|
> > > > | Antisymmetric RNN   |     0.0      |    96.0%     |  90.9%        |128              |  10K             |
> > > > | Antisymmetric RNN   |     0.0001     |    96.3%     |   91.1%       |128              |  10K             |
> > > > | Antisymmetric RNN   |      0.001      |    96.7%     |   91.2%       |128              |  10K             |
> > > > | Antisymmetric RNN   |      0.01        |    96.6%     |  90.6%        |128              |  10K             |
> > > > | Lipschitz RNN            |      -             |    *99.1%*   |  *94.2%*     |64                |  9K               |
> > > >
> > > > Results for TIMIT.
> > > >
> > > > |                                    | $\gamma$  |  MSE (test)  | N                | #parms     |
> > > > |----------------------- --|:-------------:|:-------------:|:-------------:|:--------------:|
> > > > | Antisymmetric RNN   |     0.0      |   8.03       |  420             |  200K           |
> > > > | Antisymmetric RNN   |     0.0001     |   7.76       |  420             |  200K           |
> > > > | Antisymmetric RNN   |     0.001       |   7.38       |  420             |  200K           |
> > > > | Antisymmetric RNN   |     0.01         |   7.94       |  420             |  200K           |
> > > > | Lipschitz RNN            |      -             |   *2.76*       |  256             |  200K           |
> > > >
> > > > **"If parameters $\alpha$ and $\beta$ were trainable (yet still ensuring stability) and could adapt the symmetric and anti-symmetric contributions to the problem at hand, that in my mind would be more convincing."**
> > > >
> > > > The parameters $\beta$ and $\gamma$ only affect where the optimizer is likely to end up within a finite period of time. Choosing the best tuning parameters automatically would aid implementation in a black-box context, but we rarely need to try more than a few different parameters before finding a good one. This is not the case for the antisymmetric RNN, where we have found it very difficult to tune for the diffusion term.
> > > >
> > > >
> > > > **"I also wondered whether the set of tasks used by the authors really challenges expressiveness and long-memory properties that much"**
> > > >
> > > > Here, by expressivity, we specifically refer to our increased capacity to represent more complex behaviors. By definition, we achieve this because our RNN allows for dampening/expanding behavior, unlike the antisymmetric RNN and many other projection-based methods. We consider this a separate property to memory capacity. We do achieve better performance than the antisymmetric RNN in long memory problems (e.g., the add task), but neither RNN achieves state-of-the-art at present as compared to the non-normal or exponential RNNs.

---

### Official Review · AnonReviewer1 · 2020-10-28
**CT-RNNs with stability conditions for network parametrization**

**Rating:** 6
**Confidence:** 4

**Review:**

The authors proposed a new continuous-time RNNs which appears to be extremely similar to CT-RNN (Funahashi et al. 1993). They then constraint the network representation to account for learning long-term dependencies.

Positive: The formulation of the constraint is very clear and sound.

Positive: The analysis of the stability of the model and other properties is rigor enough and to the best of my knowledge sound and correct.

Negative: From the experimental setting, reported values in tables, and code, it seems like the authors tuned the hyperparameters on the test set which I consider a bad practice that violates the code of conduct! I suspected this, and therefore, ran the code myself. With a few changes to the hyperparameters from the tuned one reported in the table the performance of the proposed model dropped significantly!

I would suggest the authors to create a fair testing scheme for all baselines and report the experimental results more accurately.

---

> ### Author Response · Authors · 2020-11-17
> **Thanks for examining and running our code.**
>
> We would like to thank the reviewer for their comments. We appreciate that the reviewer agrees that the proposed formulation of the constraint is clear and sound and that the analysis of the stability of the model and other properties is rigorous and correct. We would also like to thank the reviewer for running our provided code and verifying that a reader is able to reproduce the experiments. However, we strongly disagree with the reviewer that our experiments violate the code of ethics --- please see the discussion of our training procedure below.
>
>
> **CT-RNN formulations**
>
> There is a broad class of continuous-time RNN formulations in the theoretical literature; see our related work section and in particular Zhang et al. (2014) for a comprehensive survey of continuous-time RNNs and their stability properties. The CT-RNN model in Funahashi et al. ‘93 is of the Hopfield form, and therefore can be recast into the Cohen-Grossberg form, which we discuss in Section 3. We have provided a direct stability analysis of our particular formulation, and relate this to the Cohen-Grossberg form in Lemma 1.
>
> **Training procedure**
>
> We utilized a standard training procedure using a non-exhaustive random search within plausible ranges. We will add the ranges used in the search to the existing table of final values in the appendix. To ensure fair comparisons, we have adopted the experimental procedure in the open source implementations of published papers: For MNIST and TIMIT we adapted the code provided by [1] using the following repository https://github.com/Lezcano/expRNN. For the language modeling tasks we adapted code by [2] using the following repository https://github.com/KyleGoyette/nnRNN. For tasks such as TIMIT and the language modeling tasks, our implementation uses the provided validation set. Many accepted papers, including the implementation of [1] which we used as a baseline, often do not use a validation set for MNIST since it is known that there is no distribution shift between the training and test sets. Furthermore, we do not believe the optimization procedure is prone to overfitting as we are using a fixed stopping scheme; i.e., we reported the results of the model obtained after a fixed number of epochs dictated by a standard learning rate schedule. Nevertheless, we are quite happy to create a validation set for MNIST from the training data.
>
>
> **Sensitivity to hyperparameters**
>
> We are glad that the reviewer was able to run our implementation and reproduce the results. We welcome the reviewer to elaborate on the specific changes during the discussion period so that we may better address these. However, we are confident that our model is not any more sensitive than any other. With respect to the hyperparameters that are introduced by our architecture, Figure 3b shows a detailed ablation study for \beta showing the model performance across the range of valid values, as bounded in the derivation.
>
> **References**
>
> [1] Lezcano-Casado, M., and Martı́nez-Rubio, D. "Cheap Orthogonal Constraints in Neural Networks: A Simple Parametrization of the Orthogonal and Unitary Group." International Conference on Machine Learning. 2019.
>
> [2] Kerg, G., Goyette, K., Puelma Touzel, M., Gidel, G., Vorontsov, E., Bengio, Y., & Lajoie, G. "Non-normal Recurrent Neural Network (nnRNN): learning long time dependencies while improving expressivity with transient dynamics." Advances in Neural Information Processing Systems. 2019.

---

### Official Review · AnonReviewer3 · 2020-10-29
**Missing references - Not clear what is the improvement over existing architecture**

**Rating:** 5
**Confidence:** 4

**Review:**

I have increased my score to reflect the revisions

---------

This paper presents yet another architecture for fully connected RNNs or infinitely deep networks based on the integration of a continuous time dynamical system, where a projection of the weights is used to guarantee stability, hence a fixed point and a finite Lipschitz constant.

Positives: The maths presented in the paper is correct and their results are nicer than the considered (not exhaustive) baselines.

Negatives:

1) This idea has been widely explored and exploited by now. Adding a linear term and using a different solver is not enough in my opinion to make an innovative contribution.

If you wish to present this as an ablation study, then perhaps you need to benchmark against existing solutions.

For instance, in this (missing) reference a very similar network is presented and analysed.

@incollection{NIPS2018_7566,
title = {NAIS-Net: Stable Deep Networks from Non-Autonomous  Differential Equations},
author = {Ciccone, Marco and Gallieri, Marco and Masci, Jonathan and Osendorfer, Christian and Gomez, Faustino},
booktitle = {Advances in Neural Information Processing Systems 31},
pages = {3025--3035},
year = {2018},
publisher = {Curran Associates, Inc.},
url = {http://papers.nips.cc/paper/7566-nais-net-stable-deep-networks-from-non-autonomous-differential-equations.pdf}
}

I will refer to this as [1]. While this paper was about unrolling the stable RNN to generate a deep Lipschitz classifier, and was not used for sequence to sequence task, the architecture you propose and the claims are so much similar to [1] that this demands for a direct comparison.

In the above paper, stability projections are presented for an architecture that is essentially the same, minus the additional linear component here. Paper [1] should be included as a baseline. You have it already implemented for fully connected layers.

2) It is not very clear how this additional linear component would help in practise, as your architecture is fundamentally discretised with Euler which results in yet another generalised res-net. It has been shown that ResNet works much better than their predecessor,  Highway networks, because of the direct skip connection and better gradient flow.
While the missing reference [1] (NAIS-Net) preserves that connection, It feels like your linear term would get rid of the skip connection and prevent the technique from being used in very deep networks or very long sequences due to vanishing gradients.

3) What is the motivation for using a continuos-time approach? Your results are invalidated by the forward Euler, unless a very small hyperparameter epsilon is introduced. Why not just compute the projection in discrete time as done in [1]. Your stability will not hold for sparse in time data-points, because the Euler step would become too big and this is effectively an RNN that cannot handle different sampling time while preserving stability.

4) It seems strange to compare to NODEs as they are meant to be used for something else. In particular, NODEs are designed to work without inputs but just by estimating the initial condition for the ODE and then "unrolling".
If you add input signals to the NODE, which I guess is what you mean by "NODE RNN", how do you train it with the adjoint method? This should be made very clear in the paper.

5) Your results are limited to a fully connected architecture, while [1] has shown a method to have a Lipschitz RNN for convolutional layers. Can you generalize to that as well?

I don't feel the contribution here is relevant enough to be included in the conference.

If the above points are clarified in a convincing way and both the theoretical justification and the ablations performed with respect to [1], then I could consider improving my score.

---

> ### Author Response · Authors · 2020-11-17
> **Adding a linear term and using a different solver is not one of our main contributions, but the linear term in our model is essential.**
>
> We thank the reviewer for reading the paper and providing a detailed review. Regarding the reviewer’s summary of our work: “This paper presents yet another architecture for fully connected RNNs or infinitely deep networks based on the integration of a continuous time dynamical system, where a projection of the weights is used to guarantee stability, hence a fixed point and a finite Lipschitz constant.”
>
> * There have been many papers (indeed many published long before the reference [1]) that have investigated RNNs from the continuous-time perspective (see our references), so it seems unfair to dismiss our work based solely on this fact without mentioning our contributions. We feel that reviewer #2 gave an especially accurate summary of our work.
>
> * We emphasize that we do not consider an arbitrary projection of the weights. In particular, we argue in the text that arbitrary projections will inevitably negatively impact expressivity. Instead, we consider a parameterization that better supports stability during training while being flexible enough so that highly expressive models are attained. Our choice of parametrization makes it one of the primary features of our computational model.
>
>
> 1. **What are the innovative contributions?**
>
> We would like to stress that adding a linear term and using a different solver are not our contributions. Among others, our key contributions are the derivation of sufficient conditions for stability of our model and a novel skew-symmetric scheme that can be used to control the vanishing and exploding gradients problems. The additional linear term acts as a stabilizer, and its presence is valuable for establishing global stability criteria for the system without sacrificing expressivity.
>
> The fully connected NAIS-Net model reads $x(k+1) = x(k) + h \sigma(A x(k) + B u + b)$, and incorporates a particular projection scheme to maintain stability. In the original paper [1], this model is not formulated to deal with sequential data. (They considered whole-image tasks, as opposed to pixel-by-pixel tasks.) If we were to adapt their formulation in the obvious way to allow for sequential input (so that $x(k+1) = x(k) + h \sigma(A x(k) + B u(k) + b)$), then it would be equivalent to an Euler discretization of our general model without specific parameterization of the hidden-to-hidden matrices and without the linear term. Training this model with the same projection scheme in [1], we have not been able to achieve above 90% accuracy on pixel-by-pixel MNIST. Hence, we do not feel it is as good a baseline as LSTMs. However, if the reviewer still believes the comparison is merited, we are happy to include it for the two MNIST tasks. Also, we are happy to include a reference to the paper in the related work section.

---

> > ### Author Response · Authors · 2020-11-17
> > **Concerning the reviewer’s other comments.**
> >
> > **2. Presence of a skip connection**
> >
> > Our model takes advantage of the concept of skip connection. For instance, for the forward Euler scheme we yield the following model: $h(t+1) = h(t) + \epsilon \cdot  A h(t) + \epsilon \cdot  tanh(W h(t) + Ux(t) + b)$, where A and W is parameterized by our skew-symmetric scheme. In our experiments, the additional linear term has significant practical advantages as compared to not having this term (see our ablation study).
> >
> > **3. Benefits of continuous-time approach and discretizations**
> >
> > The motivation for using a continuous-time approach is that we can prove sufficient conditions for global stability that are independent of the specific integration scheme that is used during training (provided the step size used is sufficiently small). Our results show that a higher-order integration scheme can improve accuracy and robustness. In the case where sequential input is in the form of highly irregularly sampled data points, the Euler discretization would not be sensible and an adaptive or higher-order integrator is necessary (see [2], for example).
> >
> > **4. Neural ODE-RNNs**
> >
> > Essentially, the model described in [4] without the antisymmetric weight parameterization is a Neural-ODE RNN. Further elaborations on the idea are described in [2] and [3].  Simple Neural-ODE RNNs are difficult to train since they do not have any mechanism built in that prevents exploding gradients. We recognize that these references/definitions are missing in Section 7, so we shall add them there.
> >
> > **5. Fully connected architecture**
> >
> > Our model is not limited to a fully connected architecture. Indeed, since convolutional linear operations are simply one particular form of (stencil) matrix multiplication, one can easily apply our scheme to this setting. Our proof for global stability thus also technically applies to convolutional operations. However, we consider fully connected hidden-to-hidden matrices which are most commonly used in the related literature.
> >
> > **References**
> >
> > [1] Ciccone, Marco, et al. "Nais-net: Stable deep networks from non-autonomous differential equations." Advances in Neural Information Processing Systems. 2018.
> >
> > [2] Rubanova, Yulia, Ricky TQ Chen, and David K. Duvenaud. "Latent ordinary differential equations for irregularly-sampled time series." Advances in Neural Information Processing Systems. 2019.
> >
> > [3] Habiba, Mansura, and Barak A. Pearlmutter. "Neural Ordinary Differential Equation based Recurrent Neural Network Model." arXiv preprint arXiv:2005.09807 (2020).
> >
> > [4] Chang, B., Chen, M., Haber, E., & Chi, E. H. (2019). AntisymmetricRNN: A Dynamical System View on Recurrent Neural Networks. In International Conference on Learning Representations.

---

> > > ### Comment · AnonReviewer3 · 2020-11-23
> > > **Thanks for clarification. Where are the promised changes?**
> > >
> > > I thank the authors for clarifying some of the points raised.
> > >
> > > I still maintain that the paper addresses stability in continuous-time which is not sufficient for the RNN to be stable when discretised and it is subject to the particular discretisation method. I concede this might result in something that is more practical to train than a direct projection, however, the semidefinite parameterisation and stability in general already constrain expressivity. For example, the present paper is more restrictive than Chang.
> > > Without a direct comparison with projection-based methods, it is very hard to claim that one approach is better than another.
> > > Still, the approaches are very similar. The authors update do not reflect the analysis they proposed to do nor acknowledged the relation to the pointed reference.
> > >
> > > I'm still willing to give the authors a chance to address this point in a further revision.
> > >
> > > At the same time, while the maths could work in principle, we don't have experimental evidence that this parameterisation could work well for convolutions. So perhaps this should be pointed out as future work.

---

> > > > ### Author Response · Authors · 2020-11-23
> > > > **We will upload a revised manuscript later today.**
> > > >
> > > > We would like to thank the reviewer for taking the time to reply to our response.
> > > >
> > > > **I still maintain that the paper addresses stability in continuous-time which is not sufficient for the RNN to be stable when discretised and it is subject to the particular discretisation method.**
> > > >
> > > > Stability criteria are indeed specific to the particular discretization method, due to the addition of the time step parameter. However, true for all numerical integrators is the following statement we make under equation (1): “Given a $\beta$ and $\gamma$ that yields a globally exponentially stable continuous model, $\Delta t$ can always be chosen so that the model remains in the stability region…”. Essentially, once stability criteria for the continuous case have been obtained, one need only choose $\Delta t$ to be small enough and stability for the discretization can be guaranteed also. This is universal. However, more precise conditions on $\Delta t$ are specific to the discretization, so by analyzing the Euler method, we could not say anything about the RK2 method, or any other discretization. Therefore, by keeping to the continuous-time setting, we achieve more widespread applicability in our results.
> > > >
> > > > **I concede this might result in something that is more practical to train than a direct projection, however, the semidefinite parameterisation and stability in general already constrain expressivity.**
> > > >
> > > > In the case where $\beta = 1$ (the antisymmetric RNN with a linear term), there is an inherent limit to the type of behaviour that can be learned. Indeed, by design, any projection-based RNN trades expressivity (the capacity to represent behaviour) for trainability, to address the vanishing/exploding gradient problem. By projecting onto a subspace of the hidden-to-hidden weight matrices, one inherently limits the possible behaviours of the RNN.
> > > >
> > > > **For example, the present paper is more restrictive than Chang.**
> > > >
> > > > Our method is, by definition, a strict extension of the antisymmetric RNN of Chang et al. (again, taking $\beta = 1$ reduces to their model), so any behaviour that can be replicated by the antisymmetric RNN can be achieved with ours as well.
> > > >
> > > > **Without a direct comparison with projection-based methods, it is very hard to claim that one approach is better than another.**
> > > >
> > > > Our experiments do compare with existing projection-based RNNs. For example, we compare with Unitary RNNs, Exponential RNN, and the Antisymmetric RNN, all of which are projection-based. We cannot claim that a non-projective approach will always give better results (indeed, we know for sure that this is not the case since naively trained standard RNN models are known to have poor performance), but ours performs better in each of our experiments.
> > > >
> > > > **Still, the approaches are very similar.**
> > > >
> > > > Our model is a direct extension of the work of Chang et al. (with a linear term) so there are natural comparisons to be made. We have elaborated on the key differences in our response to Reviewer #2. Generally speaking, we achieve better results by not restricting the space of hidden-to-hidden matrices, and using a parameterization to aid training.
> > > >
> > > > **Updates that will be integrated into the revised manuscript**
> > > >
> > > > We haven’t incorporated the requested changes, since we are still running more experiments to study the performance of the (sequential) NAIS-Net. Here is what we will add tomorrow to the second revision of the manuscript:
> > > >
> > > > * Related work: `More recently, several works have adopted the dynamical systems perspective in practice to improve the stability of RNNs. For non-sequential data, Ciccone et al. (2018) proposed a negative-definite parameterization for enforcing stability in the RNN during training. Chang et al. (2018) introduced an antisymmetric hidden-to-hidden weight matrix and provided guarantees for local stability. [...]’
> > > >
> > > > * We will add an additional row to Table 1 to show that the proposed reprojection scheme by Ciccone et al. (2018) does not achieve near state of the art performance for sequential MNIST tasks:
> > > >
> > > > |                                       | ordered       | permuted     | N                | #parms        |
> > > > |----------------------------|:-------------:|:--------------:|:-------------:|:-------------:|
> > > > | Sequential NAIS-Net     |    94.3%     |  90.8%         |128              |  18K            |
> > > >
> > > > **At the same time, while the maths could work in principle, we don't have experimental evidence that this parameterisation could work well for convolutions. So perhaps this should be pointed out as future work.**
> > > >
> > > > Agreed. In theory, our parameterization directly extends to the convolutional setting, but without experimental evidence, we cannot determine if this is a good idea in practice. We have made mention of this in the conclusion as future work.

---

> > > > > ### Comment · AnonReviewer3 · 2020-11-24
> > > > > **Revised upload**
> > > > >
> > > > > I think that, after the several revisions, the paper has now significantly improved in most of the aspects highlighted by the reviewers. Although it is perhaps still not fully clear nor satisfactory regarding the experiments (for instance a conv-net would be nice to see), I will increase my score to highlight the progress made by the authors.
> > > > >
> > > > > PS: I found the removed figure 1 very nice but this is not a vital concern.

---

> > > > > > ### Author Response · Authors · 2020-11-24
> > > > > > **Thank you very much**
> > > > > >
> > > > > > We would like to thank the reviewer for the constructive feedback and dedication in reviewing our submission. We share the opinion that our manuscript and work have been significantly improved due to the proposed modifications and changes that we already incorporated, and for the further edits we plan to upload.

---

### Official Review · AnonReviewer4 · 2020-10-30
**A solid contribution**

**Rating:** 8
**Confidence:** 3

**Review:**

The paper presents some novel contributions regarding recurrent neural networks.
Building on the work of Chang et al. (2019),  the authors provide a global convergence result for the hidden representation of a family of recurrent neural networks using standard techniques from the Lyapunov analysis of dynamical systems.
The requirements of the theorem are met  (within the limits of discretization) by their proposed algorithmic scheme.
Numerical evaluation on a variety of benchmarks shows that the proposed algorithm yields systematic improvement over other RNN approaches.
For all of the above reasons, I recommend the acceptance of the paper.

Some concerns to be addressed:
- The connection between stability and trainability or refer to Chang et al. (2019) if their analysis applies here.
- specify the functions \sigma_min and \sigma_max used in Theorem 1
- specify the meaning of the one-arg function f(h^*) as opposed to the 2-arg f(h,t) appearing in Definition 1.

---

> ### Author Response · Authors · 2020-11-17
> **Thank you for the kind feedback**
>
> We thank the reviewer for this positive appraisal of our work. Regarding the mentioned issue of stability during training, the analyses of Chang et al. (2019) do not apply because we cannot guarantee (with probability one) that the process will remain stable during training. However, we will include an additional lemma that shows as $\beta \to 1$, regardless of the integration scheme, the process will remain stable with higher probability during training, provided the initial parameters lie within the stable region. For more details on this, please see our response to Reviewer #2.
>
> Regarding the other comments:
> - Definitions for sigma_min and sigma_max as minimum and maximum spectral values will be provided in the updated manuscript.
> - The one-arg function $f(h^*)$ in Definition 1 is a typo, thank you for pointing this out. It should read $f(h^*, t) = 0$.

---

### Author Response · Authors · 2020-11-17
**General response**

We would like to thank all the reviewers for the detailed and constructive responses, and in particular, R2 and R4 for the positive feedback.

We stress that we do not claim as a contribution the particular algebraic form of a linear term plus a nonlinear term. First, we chose a particular abstract system of differential equations (Eq. (1a)) that had appealing properties from a dynamical systems point of view and in turn it allowed us to study global stability of the model using tools from control theory. The additional linear term acts as a stabilizer, and its presence is valuable for establishing global stability criteria for the system without sacrificing expressivity. The derived sufficient conditions for stability motivated a novel scheme for parameterizing the hidden-to-hidden matrices. These insights led to a novel skew-symmetric scheme for parameterizing the hidden-to-hidden weights.

Incorporating the feedback from all reviewers, we will make the following changes to the text:

* We will expand Eq. 1 from the general form by plugging-in our symmetric/skew-symmetric parameterization. Expanding, the architecture takes the form $$\frac{dh(t)}{dt} = ((1-\beta) (Q-Q^T) + \beta (Q+Q^T) - \gamma I)h(t) + \tanh( ((1-\beta) (P-P^T) + \beta (P+P^T) - \gamma I)h(t) + Ux(t) + b).$$ This makes the final computational graph easier for the reader to see, and makes it clear that this is our major contribution within this class of models.

* In response to constructive comments by R2, we will prove an additional lemma which shows that the rate of change in the eigenvalues of the symmetric part of the parameterized hidden-to-hidden weight matrices during gradient descent diminishes quadratically as $\beta \to 1$. This implies that for choices of beta close to one (which tends to be the case), the training process is likely to remain in the stability region with higher probability during training, provided it starts there. We demonstrate this property empirically through additional figures which show small changes in these eigenvalues during training.

* We will add standard deviations in our experimental results tables from multiple starts.

* The range of hyperparameters used in the  grid search will be included in the hyperparameter tables in the appendix.

---

### Decision · Program_Chairs · 2021-01-07
**Final Decision**

**Decision:**

Accept (Poster)

**Comment:**

Solid work on extending AntisymmetricRNN and expanding its expressivity while controlling the global stability of the recurrent dynamics. It contributes to the growing interest in continuous-time RNN formulations that can deal with exploding gradient problem, and worthy of ICLR poster presentation. Three reviewers were positive and one was slightly negative. Authors added additional experiments and strengthened the manuscript significantly during the review process.